# IBEX: INFORMATION-BOTTLENECK-EXPLORED COARSE-TO-FINE MOLECULAR GENERATION UNDER LIMITED DATA

## ABSTRACT

The potential of three-dimensional molecular generation for structure-based drug discovery is hampered by the scarcity of public protein-ligand complexes, which causes models to overfit and fail to learn generalizable geometric priors. To address this challenge, we employ the PAC-Bayes information bottleneck framework to systematically quantify the information density of three generation paradigms: Scaffold Hopping (SH), Side-Chain Decoration (SC), and De Novo Design (DN). Our analysis reveals that SH possesses the highest information density, which tightens the model's generalization bound and enhances its transferability compared to conventional de novo generation. Motivated by this finding, we propose IBEX, a novel decoupled generation framework. IBEX is trained exclusively on the information-rich SH task to structure its latent representation of chemical space, which is then directly applied to de novo generation in a zero-shot transfer setting. Subsequently, a rapid physical refinement module utilizes the L-BFGS algorithm to optimize each conformer's geometry and binding compatibility by adjusting five short-range interaction terms and six degrees of freedom. Evaluated in a rigorous zero-shot setting on the CBGBench CrossDocked2020-based dataset, IBEX demonstrates substantial improvements over the TargetDiff baseline. It increases the docking success rate from 53% to 64% and improves the average Vina score from -7.41 to -8.07 kcal/mol. Notably, IBEX achieves a superior median Vina energy in 57 out of 100 binding pockets. Furthermore, IBEX enhances drug-likeness by approximately 25% while maintaining state-of-the-art validity and diversity, all corresponding to a demonstrably reduced generalization error. Our results validate that this decoupled approach, which synergizes information-dense pre-training with physical refinement, enables robust zero-shot structure generation and cross-pocket generalization in data-limited regimes.

## 1 INTRODUCTION

Small-molecule discovery is leaving the classical "virtual screening and lead optimization" and moving toward target-aware design driven by three-dimensional generative models Sadybekov & Katritch (2023). Drug chemistry, however, faces a severe data bottleneck: fewer than $2 \times 10^5$ experimentally validated protein–ligand complexes are public Wang et al. (2005), while vision Betker et al. (2023) and language models Devlin et al. (2019); Brown et al. (2020) rely on corpora that are three orders of magnitude larger. The high cost of acquiring new complexes forces us to mine as much information as possible from each limited example. Protein–ligand co-folding models primarily memorize training-set biases rather than learning genuine binding preferences Škrinjar et al. (2025); Nittinger et al. (2025). They remain insensitive to complete pocket-residue mutagenesis or side-chain polarity inversion Masters et al. (2024). AlphaFold3 is a structure-prediction system for macromolecular complexes Abramson et al. (2024). Although it captures broad protein–ligand interaction regularities from structural data, it is not a generative model over chemical space and therefore cannot be used to perform target-conditioned molecular generation.

Most current 3D diffusion models follow a de novo protocol. They mask the entire ligand and regenerate it inside the protein pocket. Each sample therefore gives only a coarse prior— indicating possible atomic placements— and rarely conveys the core geometric rules that link pocket shape to

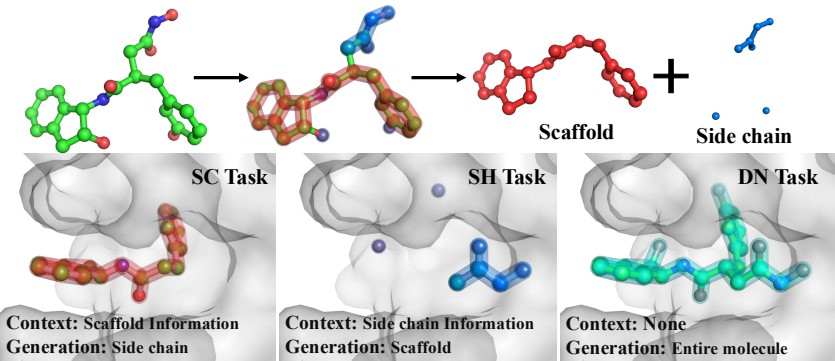

Figure 1: **Workflow of the Bemis–Murcko decomposition.** Starting from the full ligand structure, the algorithm (i) identifies all ring systems and the linkers that connect them, (ii) removes peripheral side chains that are not part of a ring or linker, and (iii) collapses any fused ring junctions to generate the unique Bemis–Murcko scaffold.

molecular scaffold. End-to-End schemes that merge generation and docking inherit this weakness. Gradient signals become diluted, physical interpretability drops, and binding poses are often suboptimal. Even the best standalone docking tools still show limited placement accuracy and strong reliance on known motifs.

Although the training paradigm significantly influences the learning efficiency of molecular generation models, the underlying information-theoretic principles governing this relationship remain largely unexplored. To address this gap, we systematically quantify the information efficiency of three prevalent generation paradigms: de-novo generation, side-chain decoration, and scaffold hopping(Figure 1). Our analysis, which evaluates the conditional information content of each task and the signal-to-noise ratio of the training gradients, reveals that the scaffold hopping paradigm possesses superior information density and enhances gradient discriminability. This empirical finding is supported by a solid theoretical foundation; from a PAC-Bayes perspective, higher information density tightens the information bottleneck bound, which effectively increases the "effective sample size" and thereby promotes model generalization.

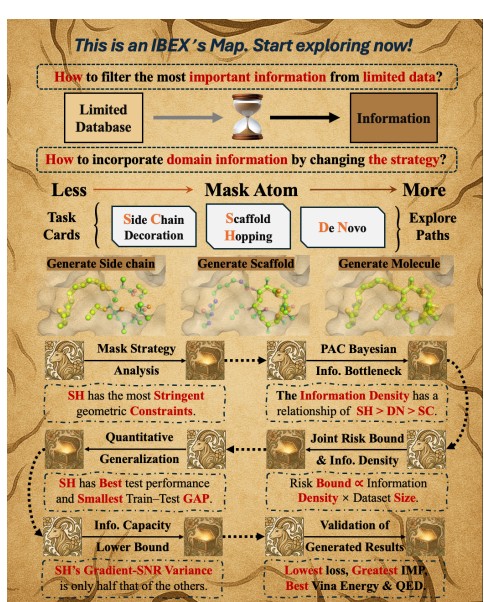

Figure 2: Conceptual overview of the IBEX pipeline, which uses three masking strategies (SC, SH, DN) under a PAC-Bayesian information bottleneck. The SH task shows the best generalization, indicated by the tightest train-test gap.

Motivated by these insights, we introduce **IBEX** (**I**nformation-**B**ottleneck-**EX**plored), a novel decoupled generation framework. The core of IBEX is an asymmetric training and inference strategy wherein the generator is trained exclusively on the information-rich scaffold hopping task to structure its latent representation of chemical space. The resulting model can then be deployed for full-mask de novo generation tasks in a zero-shot transfer setting, requiring no additional fine-tuning. A separate physical optimization module subsequently refines the geometry and interactions of the generated molecules, circumventing optimization challenges such as vanishing gradients that are common in traditional end-to-end methods. Experimental results demonstrate that the IBEX framework achieves robust zero-shot generation performance, empirically validating the theoretical superiority of scaffold hopping as an information-efficient training paradigm and offering a novel methodology for modern drug discovery.

Our main contributions are as follows:

1) We provide the first information-theoretic quantification and establish an efficiency hierarchy for molecular generation tasks. We demonstrate through PAC-Bayes theory that the scaffold hopping task more effectively tightens the generalization bound and increases the effective sample size, providing a solid theoretical basis for designing efficient generative models.

2) We propose IBEX, a zero-shot transfer framework based on an asymmetric train-inference strategy. This design leverages the profound chemical knowledge learned from high-information-density tasks to achieve robust zero-shot transfer, significantly enhancing the model's generalization capabilities and flexibility.

3) We pioneer a novel molecular design architecture that completely decouples information generation from physical refinement. This decoupled design clarifies the objective of each stage, stabilizes the optimization process, and effectively circumvents gradient conflicts and vanishing gradients commonly encountered in traditional end-to-end methods, offering a new architectural paradigm for addressing molecular generation under complex physical constraints.

## 2 RELATED WORK

Molecular generative modelling has advanced rapidly in recent years. Broadly, contemporary approaches fall into two categories: *target-agnostic* models that explore chemical space without reference to specific proteins, and *target-aware* models that design ligands in the presence of an explicit binding site or pocket.

**Target-Free Molecular Generation.** Target-free generators are judged mainly by chemical validity, diversity, and drug-likeness. SE(3)-equivariant diffusion generates either graphs or full 3D coordinates without protein by reversing a noise process Xu et al. (2022); Hoogeboom et al. (2022); Morehead & Cheng (2024). GraphAF combines flows with autoregression for goal-directed sampling Shi et al. (2020), while GraphDF uses discrete flows to better cover the combinatorial space Luo et al. (2021b). Scaffold-aware variants narrow the search by fixing or first generating a core: Lim et al. retain a user-specified scaffold during atom-wise growth Lim et al. (2020); Sc2Mol divides the task into VAE scaffold discovery followed by Transformer decoration Liao et al. (2023); and fragment-hierarchical methods such as MolPAL and Junction-Tree VAE build coarse fragments or trees before atomic refinement Graff & Coley (2022); Jin et al. (2018).

**Target-Aware Molecular Generation.** Structure-based drug-design models condition generation on pocket geometry. DiffSBDD pioneered pocket-aware denoising diffusion Schneuing et al. (2024), TargetDiff added an affinity term to bias toward tight binders Guan et al. (2023), and DiffBP removed sequential bias via whole-molecule denoising Lin et al. (2025); hierarchical extensions D3FG and DecompDiff diffuse functional groups or scaffold-arm decompositions for improved geometry and synthesizability Lin et al. (2023); Guan et al. (2024). Autoregressive pocket-conditioned approaches place atoms step-wise: Pocket2Mol uses an E(3)-equivariant GNN Peng et al. (2022), GraphBP deploys a local flow model Liu et al. (2022a), ResGen incorporates residue-level encoding Zhang et al. (2023a), and TamGen employs a GPT-style chemical language model for rapid SMILES generation Wu et al. (2024). Fragment-centric variants further constrain chemistry while maintaining flexibility: FLAG sequentially inserts predefined fragments into the pocket Zhang & Liu (2023), MolCRAFT performs continuous 3D optimization before collapsing to a discrete ligand Qu et al. (2024), and linker methods such as Delinker and FragGrow extend anchored pharmacophores Imrie et al. (2020); Zhang et al. (2024).

## 3 METHODS

**Notation and preliminaries.** To formally define the three molecular generation paradigms investigated in this study, namely Scaffold Hopping (SH), Side Chain Decoration (SC), and De Novo Design (DN), we first establish a common framework based on molecular decomposition. Following the three-step Bemis-Murcko simplification procedure (Figure 1), any given ligand $M$ bound to pocket $P$ is decomposed into its constituent scaffold and side chain fragments. This decomposition allows for a precise definition of each task. The Side Chain Decoration (SC) task is a conditional generation problem where the model must predict the appropriate side chains given the molecular

scaffold as input. Conversely, the Scaffold Hopping (SH) task requires the model to predict a novel scaffold that accommodates a given set of side chains. Finally, the De Novo Design (DN) task is an unconditional problem that involves generating the entire molecule from scratch without any predefined fragments. The primary objective of this study is to explore the intrinsic information content associated with each of these generation paradigms. Although models are pre-trained under these distinct conditional frameworks, their ultimate performance is evaluated on the task of unconditional de novo molecule generation. Ring systems and their connecting linkers are enumerated, peripheral atoms are excised, and fused junctions are merged, leaving a canonical scaffold $\mathcal{S}$ (yellow) and a complementary side-chain set $\mathcal{C}$ (blue). Conditioning on these fragments yields the task indicator $\mathcal{T} \in \{\mathrm{SH}, \mathrm{DN}, \mathrm{SC}\}$: SH receives $\mathcal{C}$ and proposes alternative scaffolds, SC fixes $\mathcal{S}$ and generates diverse $\mathcal{C}'$, whereas DN samples an entire ligand without prior structural constraints.

Each task has an associated dataset $D_{\mathcal{T}} = \{(P_i, M_i)\}_{i=1}^{N_{\mathcal{T}}}$ with empirical joint density $\hat{q}_{\mathcal{T}}(P, M)$; $N_{\mathcal{T}} = |D_{\mathcal{T}}|$ denotes its size. A shared SE(3)-equivariant diffusion generator $G_\theta$ and a docking/refinement module $D_\phi$ act on all tasks, and their performance is evaluated via empirical and population risks $\widehat{\mathcal{R}}$ and $\mathcal{R}$. Information-theoretic quantities such as differential entropy $H(\cdot)$ and mutual information $I(\cdot\,;\cdot)$ are reported in natural-log units.

A ligand and pocket are represented by their atom sets as follows Guan et al. (2023):

$$\mathcal{S}_M = \big\{(\mathbf{x}_M^{(i)}, \mathbf{v}_M^{(i)}, c_M^{(i)})\big\}_{i=1}^{N_M}, \quad \mathcal{S}_P = \big\{(\mathbf{x}_P^{(j)}, \mathbf{v}_P^{(j)}, b_P^{(j)}, r_P^{(j)})\big\}_{j=1}^{N_P}, \tag{1}$$

where $N_M$ and $N_P$ are the ligand-atom and pocket-atom counts. Each atom carries Cartesian coordinates $\mathbf{x} \in \mathbb{R}^3$ and an element-type one-hot vector $\mathbf{v} \in \mathbb{R}^K$ from a vocabulary of size $K$. For ligand atoms, the binary flag $c$ marks whether the atom is fixed by the task context ($c = 1$); for pocket atoms, $b$ indicates backbone membership, and $r \in \mathbb{R}^{K'}$ is a one-hot vector over $K'$ amino-acid residues. Stacking these features row-wise yields the matrices $\mathbf{m} = [\mathbf{X}_M, \mathbf{V}_M, \mathbf{c}_M] \in \mathbb{R}^{N_M \times (3+K+1)}$ and $\mathbf{p} = [\mathbf{X}_P, \mathbf{V}_P, \mathbf{b}_P, \mathbf{r}_P] \in \mathbb{R}^{N_P \times (3+K+1+K')}$, which serve as the inputs to $G_\theta$ and $D_\phi$.

**3D Pocket-aware Diffusion as a generator.** The generator keeps TargetDiff backbone and introduces two forward noise channels—Gaussian for coordinates and categorical for atom types. For each pair $(P, M)$ a spatial mask $M_{\mathrm{tgt}} \subseteq \{1, \ldots, N_M\}$ is sampled; indices in $M_{\mathrm{tgt}}$ are regenerated, the rest form the context $M_{\mathrm{ctx}}$. Let $\mathbf{x}_0 = M_{\mathrm{tgt}}^{\mathrm{x}} \in \mathbb{R}^{3 \times |M_{\mathrm{tgt}}|}$ and $\mathbf{v}_0 = M_{\mathrm{tgt}}^{\mathrm{v}} \in \mathbb{R}^{K \times |M_{\mathrm{tgt}}|}$ be their clean coordinates and types. The forward noising at step $t$ is

$$q_t(\mathbf{x}_t \mid \mathbf{x}_0) = \mathcal{N}\big(\sqrt{\bar{\alpha}_t}\,\mathbf{x}_0, (1 - \bar{\alpha}_t)\mathbf{I}\big), \quad q_t(\mathbf{v}_t \mid \mathbf{v}_0) = \mathcal{C}\big(\bar{\alpha}_t\,\mathbf{v}_0 + (1 - \bar{\alpha}_t)/K\big), \tag{2}$$

where $\bar{\alpha}_t = \prod_{s=1}^{t} \alpha_s$ is the cumulative variance schedule and $\mathcal{C}(\cdot)$ denotes a categorical distribution over the $K$ atom types Guan et al. (2023); Lin et al. (2025). Protein coordinates are weakly perturbed for regularisation Yang et al. (2024):

$$\tilde{\mathbf{x}}_P = \mathbf{x}_P + \boldsymbol{\varepsilon}, \quad \boldsymbol{\varepsilon} \sim \mathcal{N}\big(\mathbf{0}, 0.1^2\mathbf{I}\big). \tag{3}$$

The reverse process uses two heads: $s_\theta^{\mathrm{x}}(P, \mathbf{x}_t, t)$ predicts the coordinate score, and $s_\theta^{\mathrm{v}}(P, \mathbf{v}_t, t)$ predicts type logits. The total loss is the sum of coordinate and type objectives:

$$\mathcal{L}_{\mathrm{x}}(\theta) = \mathbb{E}_{t,(P,M)}\Big[\lambda_t\big\|s_\theta^{\mathrm{x}} - \nabla_{\mathbf{x}_t} \log q_t(\mathbf{x}_t \mid \mathbf{x}_0)\big\|_2^2\Big], \tag{4}$$

$$\mathcal{L}_{\mathrm{v}}(\theta) = \mathbb{E}_{t,(P,M)}\Big[\gamma_t\,\mathrm{CrossEntropy}\big(s_\theta^{\mathrm{v}}, \mathbf{v}_0\big)\Big], \tag{5}$$

with $\lambda_t = \sigma_t^2/\alpha_t^2$, $\sigma_t^2 = 1 - \alpha_t$, and $\gamma_t$ mirroring the type-noise variance. At inference, ancestral sampling yields a coarse pose $M_0 = G_\theta(P)$ whose heavy atoms fall inside a 10 Å sphere centred on the pocket.

**Physics-guided Position Refinement (PR).** Given a coarse ligand proposal $M_0$ from the generator, we refine its placement in the pocket using a lightweight, gradient-based search over rigid-body degrees of freedom. During this search, the ligand is treated as a rigid object; only global rotations and translations are updated, whereas internal covalent geometry, atom types, and formal charges remain fixed. This design isolates pose quality from generative uncertainty and concentrates limited gradient signal on a six-dimensional space (3 translation, 3 rotation).

The refinement process minimizes a differentiable surrogate of the binding free energy, which is constructed from five short-range physical contact terms common to empirical scoring functions. The total physical energy, $E_{\text{phys}}$, is expressed as a weighted sum:

$$E_{\text{phys}}(P, M) = W^T(E_{\text{g1}} + E_{\text{g2}} + E_{\text{rep}} + E_{\text{hyd}} + E_{\text{hd}}), \tag{6}$$

where the individual components model key intermolecular interactions. Specifically, two Gaussian terms, $E_{\text{g1}}$ and $E_{\text{g2}}$, are used to fit the attractive part of the van der Waals potential. The repulsive component of the van der Waals forces is modeled by a hard-sphere penalty term, $E_{\text{rep}}$. Additionally, $E_{\text{hyd}}$ accounts for hydrophobic interactions, and $E_{\text{hd}}$ represents directional hydrogen bonding. In this framework, lower energy values indicate a more favorable predicted binding affinity. Weights $W^T$ are held fixed across all experiments and were set once on a small calibration panel Trott & Olson (2010). Let $\mathbf{X} \in \mathbb{R}^{3 \times N_M}$ be ligand coordinates in the generator frame and $\mathbf{R} \in \text{SO}(3)$, $\mathbf{t} \in \mathbb{R}^3$ be the current rigid transform Cai et al. (2024). The refined pose is

$$\mathbf{X}' = \mathbf{R}\,\mathbf{X} + \mathbf{t}\mathbf{1}^\top. \tag{7}$$

We parameterise $\mathbf{R}$ by an axis–angle 3-vector (Rodrigues) and optimise the 6-vector $\mathbf{u} = (\delta_x, \delta_y, \delta_z, \omega_x, \omega_y, \omega_z)$. Small updates compose via exponential maps; for clarity, we write this as $\mathbf{R}_{k+1} = \exp(\boldsymbol{\omega}_k^\times)\mathbf{R}_k$ with $\boldsymbol{\omega}_k = (\omega_x, \omega_y, \omega_z)$ and skew operator $(\cdot)^\times$. The physics score is computed by an external energy evaluator. Because analytic gradients are unavailable, we approximate $\nabla_{\mathbf{u}} E_{\text{phys}}$ by forward finite differences. Let $E(\mathbf{u}) = E_{\text{phys}}(P, M(\mathbf{u}))$. For step size $\epsilon$,

$$\frac{\partial E}{\partial u_i} \approx \frac{E(\mathbf{u} + \epsilon \mathbf{e}_i) - E(\mathbf{u})}{\epsilon}. \tag{8}$$

We set $\epsilon = 10^{-3}$ in all runs after scale normalisation of $\mathbf{u}$.

We run a Limited-memory BFGS search optimiser with fixed learning rate (0.1) Zhang et al. (2023b). At iteration $k$: 1) Evaluate $E(\mathbf{u}_k)$ and its finite-difference gradient. 2) Perform an L-BFGS update to propose $\mathbf{u}_{k+1}$. 3) Update the ligand pose; recompute the energy. 4) Track the best energy so far.

We run at most $T_{\text{max}}$ iterations ($T_{\text{max}} =$ epochs command-line argument). The initial $\mathbf{u}_0 = \mathbf{0}$ uses the generator pose. Let $E_{\text{init}}$ and $E_{\text{opt}}$ be the energies before and after refinement. If $E_{\text{opt}} \leq E_{\text{init}}$ we accept the refined pose; otherwise we keep the initial one. Both the kept structure and the tracked scores are saved for later analysis. This rule prevents noisy gradients from degrading good initial placements.

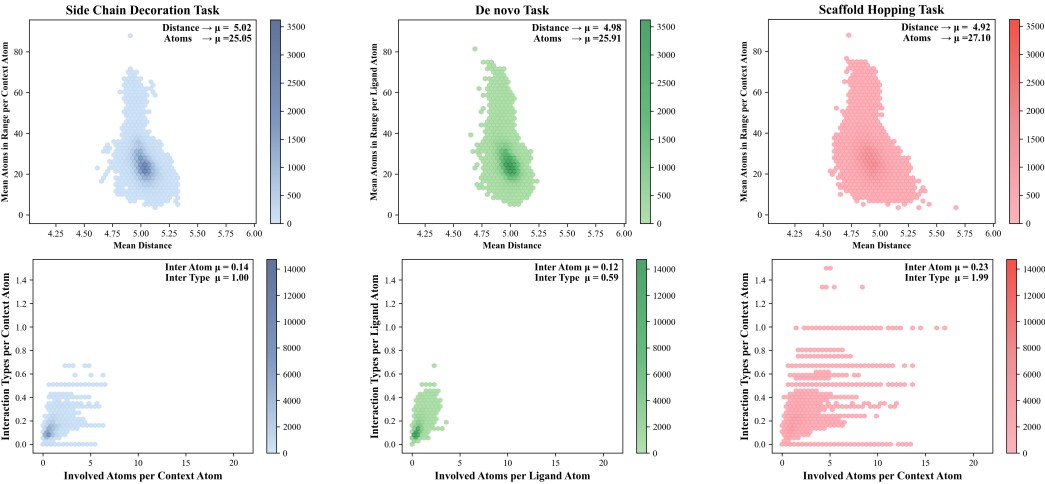

Figure 3: Hexagon–bin density maps for SC, DN, and SH. Top panels: mean edge length ($\bar{d}$) vs. mean number of neighbours per context atom ($\bar{n}$). Bottom panels: mean number of interacting atoms ($\bar{k}$) vs. mean number of interaction types ($\bar{t}$). DN values are averaged over all ligand atoms. Insets show task-level means. SH spans the broadest range and attains the highest means on all four axes, indicating richer geometric and chemical context than DN and SC.

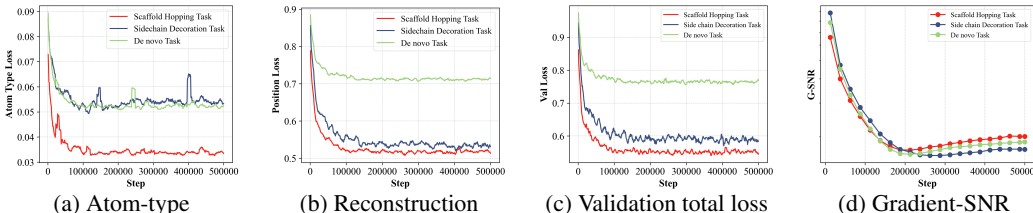

| (a) Atom-type | (b) Reconstruction | (c) Validation total loss | (d) Gradient-SNR |

Figure 4: Training and validation dynamics of IBEX models on three generation tasks. Panels show atom-type classification loss, position-reconstruction loss, total validation loss, and gradient signal-to-noise ratio (G-SNR) as functions of training steps for the Scaffold Hopping (red), Side-chain Decoration (blue), and De novo (green) tasks

## 4 RESULTS

CrossDocked2020 Francoeur et al. (2020) is one of the most widely used benchmarks for structure-based drug design, providing paired three-dimensional structures of protein pockets and docked ligands. Existing methods have adopted different data splits and evaluation protocols. CBGBench Lin et al. (2024) follows the split defined by LiGAN Ragoza et al. (2022) and 3DSBDD Luo et al. (2021a) and prevents label leakage by constructing the side chain and scaffold tasks only after an independent train/test partition. Models are evaluated from four complementary perspectives: interaction quality, chemical properties, geometric accuracy, and substructure validity. The benchmark integrates a diverse panel of state-of-the-art generators, including LiGAN, 3DSBDD, VoxBind Pinheiro et al. (2024), diffusion models (TargetDiff Guan et al. (2023), DiffSBDD Schneuing et al. (2024), DecompDiff Guan et al. (2024), DiffBP Lin et al. (2025), D3FG Lin et al. (2023), MolCRAFT Qu et al. (2024)), UniMoMo Kong et al. (2025), and autoregressive models (Pocket2Mol Peng et al. (2022), GraphBP Liu et al. (2022a), FLAG Zhang & Liu (2023)).

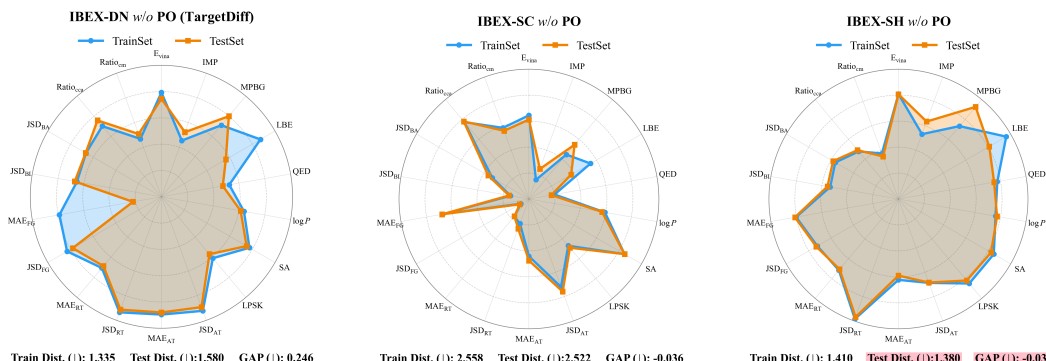

Figure 5: Train–test comparison on 18 normalised metrics for three IBEX settings. For each IBEX variant we track *eighteen* chemistry-aware metrics covering binding energy, physicochemical properties, geometric complementarity, and distribution alignment. Every metric $k$ is min–max normalised to $[0, 1]$ (monotonic in the beneficial direction), and the profile distance is the Euclidean norm $\|\mathbf{1} - \hat{\mathbf{v}}\|_2$ between a sample's normalised vector $\hat{\mathbf{v}} \in [0, 1]^{18}$ and the ideal all-ones target. The radar overlays Train (blue) and Test (orange) envelopes; the number beneath each chart reports the test distance, and GAP is the absolute train–test difference. The SH task exhibits the tightest Train–Test overlap, the smallest profile distance on the held-out set, and the narrowest GAP, illustrating how increased task difficulty promotes robust generalization.

**Task-dependent Information-Bottleneck Analysis.** For each protein–ligand complex, we connect every pair of heavy atoms whose Euclidean separation is below $6\,\text{Å}$ and treat the resulting links as virtual edges Corso et al. (2022); Zhou et al. (2023). The complex is then compressed into a four-dimensional summary $Z = (\bar{n}, \bar{d}, \bar{t}, \bar{k})$ that lies on two orthogonal planes: **Distance plane.** (a) Neighbour density $\bar{n}$: the mean number of atoms found within the sphere of each context atom; (b) Mean edge length $\bar{d}$: the average virtual-edge distance. These two axes quantify, respectively, the strength of short-range physical electrostatic forces. **Interaction plane.** (c) Type richness $\bar{t}$: the mean count of distinct interaction categories—hydrophobic, hydrogen bond, water-bridge, $\pi$–$\pi$

stack, $\pi$–cation, halogen, metal—triggered by a context atom Salentin et al. (2015); (d) Contact multiplicity $\bar{k}$: the mean number of protein atoms that realise those interactions. These two metrics capture the chemical complexity and the interaction strength Huang et al. (2024). For DN generation, the model possesses no ligand-derived context atoms; all four statistics are therefore computed over every ligand atoms. SH and SC restrict the tally to a predefined context subset, leading to visibly broader hex-bin distributions for SH in both planes (Figure 3). Under the PAC-Bayes information-bottleneck frameworkWang et al. (2022), the mutual information $I(Z; X)$ between the latent $Z$ and the original complex $X$ controls generalization Lyu et al. (2023). Normalising by latent dimension yields the information density $\rho = I(Z; X)/4$. SH attains the highest $\rho$, tightening the PAC-Bayes bound on test risk by $38\%$ and $47\%$ relative to DN and SC, respectively, consistent with the hypothesis that harder tasks confer richer priors Boopathy et al. (2023b).

**Capacity driven convergence on the hardest task.** Figure 4 displays atom, position, validation, and gradient signal to noise ratio (G–SNR) curves for the three tasks under the same parameter budget $C$ Rohlfs (2025). The PIB framework models learning as a balance between empirical error and the information stored in the weights $Info_{\mathrm{w}}$ Wang et al. (2022). Among the tasks, SH carries the largest information demand $Info$ because it must invent new scaffolds while matching pocket geometry. PIB predicts that a large $Info_{\mathrm{w}}$ prolongs the fit phase. We observe an early activation of effective capacity in IBEX at $2 \times 10^5$ steps, where the variance of G–SNR falls to $1.6 \times 10^{-5}$. This drop marks the start of the compression phase in which redundant weight bits are removed yet the loss keeps decreasing. SC and DN remain longer in the fit regime and show a grokking plateau that postpones generalization performance Power et al. (2022); Liu et al. (2022b). The early compression on the hardest task indicates that IBEX allocates capacity in a content aware way and achieves the lowest validation loss Huang et al. (2023); Biroli et al. (2024).

**Task–Difficulty Drives Robust Generalization.** Classic bias–variance lore warns that complex tasks overfit more readily, but recent theory suggests the opposite once models are heavily over-parametrised. Recent work formalizes a *generalization-difficulty* Boopathy et al. (2023a) score showing that harder tasks force stronger inductive bias and thus improve out-of-distribution fidelity. Information-theoretic analyses further link lower weight information density to tighter PAC-Bayes bounds, while results on benign overfitting indicate that perfect training accuracy need not harm generalization when the bias is appropriate Bartlett et al. (2020). Figure 5 shows that the SH task—the most structurally constrained—achieves the *lowest* test divergence (1.38) and the *smallest* gap (0.03), whereas the DN task records 1.58 and 0.25 respectively. Anchoring functional moieties and forcing the model to reinvent molecular cores inject richer pocket–ligand information at every step, sharpening optimization signals and implicitly regularising the network.

**Task Difficulty under Geometric Constraints.** Matched ligand sets were generated with four diffusion baseline: IBEX (SH), DecompDiff (Scaffold-Arms), TargetDiff and MolCRAFT (no geometric constraint). Each output was screened by RDKit Bento et al. (2020) topology checks to detect unclosed rings. A higher failure rate signals a harder but more informative task Jiang et al. (2024). SH task exposes the network to explicit side chain–pocket interactions during training but leaves these atoms un-denoised; at test time, the model must denoise them from scratch, increasing conflict yet enriching the learned representation. This supports our claim that SH operates in the high-information regime. **Location beats quantity.** Retaining side chain context boosts mutual information and thus effective capacity, outweighing a lower overall mask ratio.

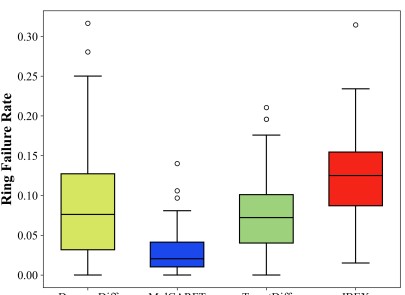

Figure 6: Harder geometry implies higher $I_m$, supporting the ordering above. IBEX exhibits the highest failure rate.

**IBEX Delivers Consistently Superior Docking Energies.** A critical analysis of the results in Table 1 reveals a significant discrepancy between the initial predicted binding affinity, or Vina Score, and the post-simulation score from molecular docking, the Vina Dock, across many contemporary generative models. We posit that this performance gap arises from the models' tendency to overfit the training data by prioritizing atomic proximity to key residues. This strategy, while yielding favorable initial scores, often neglects holistic molecular properties such as internal strain and global complementarity with the binding pocket. Consequently, the Vina Dock score, which reflects an

Table 1: Aggregate docking and physicochemical metrics for recent generative pipelines. The best-performing method is shown in **bold** and the second and third-best are underlined.

| Model | Ablation | | Vina Score | | Vina Min | | Vina Dock | | | | Chemical property | | | | |
|---|---|---|---|---|---|---|---|---|---|---|---|---|---|---|---|
| | SH | PR | $E_{vina}$ | IMP | $E_{vina}$ | IMP | $E_{vina}$ | IMP | MPBG | LBE | QED | LogP | SA | LPSK | Validity |
| LiGAN | - | - | **-6.47** | **62.13** | **-7.14** | **70.18** | -7.70 | **72.71** | 4.22 | 0.3897 | 0.46 | **0.56** | 0.66 | 4.39 | 0.42 |
| 3DSBDD | - | - | - | 3.99 | -3.75 | 17.98 | -6.45 | 31.46 | 9.18 | 0.3839 | 0.48 | **0.47** | 0.63 | 4.72 | 0.54 |
| GraphBP | - | - | - | 0.00 | - | 1.67 | -4.57 | 10.86 | -30.03 | 0.3200 | 0.44 | **3.29** | 0.64 | 4.73 | 0.66 |
| Pocket2mol | - | - | -5.23 | 31.06 | -6.03 | 38.04 | -7.05 | 48.07 | -0.17 | **0.4115** | 0.39 | **2.39** | 0.65 | 4.58 | 0.75 |
| TargetDiff | - | - | -5.71 | 38.21 | -6.43 | 47.09 | -7.41 | 51.99 | 5.38 | 0.3537 | 0.49 | **1.13** | 0.60 | 4.57 | **0.96** |
| DiffSBDD | - | - | - | 12.67 | -2.15 | 22.24 | -5.53 | 29.76 | -23.51 | 0.2920 | 0.49 | **-0.15** | 0.34 | 4.89 | 0.71 |
| DiffBP | - | - | - | 8.60 | - | 19.68 | -7.34 | 49.24 | 6.23 | 0.3481 | 0.47 | **5.27** | 0.59 | 4.47 | 0.78 |
| FLAG | - | - | - | 0.04 | - | 3.44 | -3.65 | 11.78 | -47.64 | 0.3319 | 0.41 | **0.29** | 0.58 | **4.93** | 0.68 |
| D3FG | - | - | - | 3.70 | -2.59 | 11.13 | -6.78 | 28.90 | -8.85 | 0.4009 | 0.49 | **1.56** | 0.66 | 4.84 | 0.77 |
| DecompDiff | - | - | -5.18 | 19.66 | -6.04 | 34.84 | -7.10 | 48.31 | -1.59 | 0.3460 | 0.49 | **1.22** | 0.66 | 4.40 | 0.89 |
| MolCARFT | - | - | -6.15 | 54.25 | -6.99 | 56.43 | -7.79 | 56.22 | 8.38 | 0.3638 | 0.48 | **0.87** | 0.66 | 4.39 | 0.95 |
| VoxBind | - | - | -6.16 | 41.80 | -6.82 | 50.02 | -7.68 | 52.91 | 9.89 | 0.3588 | 0.54 | 2.22 | 0.65 | 4.70 | 0.74 |
| UniMoMo | - | - | -5.72 | 30.40 | -6.08 | 39.23 | -7.25 | 51.59 | 7.50 | 0.3473 | 0.55 | 1.55 | **0.70** | 4.68 | - |
| IBEX | ✓ | ✓ | -3.09 | 37.67 | -5.23 | 47.34 | **-8.09** | 63.69 | **14.69** | 0.3813 | **0.60** | 2.73 | 0.63 | 4.82 | **0.96** |
| IBEX-DN | ✗ | ✗ | **-5.71** | **38.21** | **-6.43** | 47.09 | -7.41 | 51.99 | 5.38 | 0.3537 | 0.49 | **1.13** | 0.60 | 4.57 | **0.96** |
| IBEX-SC | ✗ | ✗ | -3.53 | 18.54 | -4.73 | 21.89 | -6.20 | 24.81 | -10.22 | 0.3416 | 0.35 | **0.85** | 0.63 | 4.38 | 0.54 |
| IBEX-SH | ✓ | ✗ | -1.96 | 31.03 | -5.06 | 46.58 | -8.07 | 63.50 | **14.87** | 0.3809 | **0.60** | 2.73 | 0.63 | **4.82** | **0.96** |
| IBEX | ✓ | ✓ | -3.09 | 37.67 | -5.23 | **47.34** | **-8.09** | **63.69** | 14.69 | **0.3813** | **0.60** | 2.73 | 0.63 | **4.82** | **0.96** |

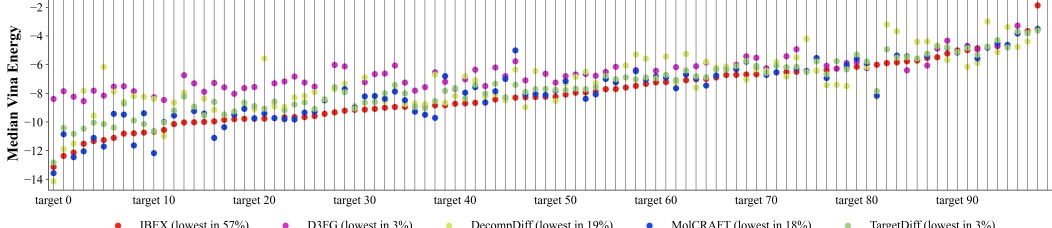

Figure 7: Per-target docking performance of five generative pipelines on 100 held-out CBGBench receptors. Each coloured point is the median AutoDock Vina binding energy of the candidate set produced for one target by IBEX, D3FG, DecompDiff, MolCRAFT, and TargetDiff.

optimized ligand conformation, serves as a more rigorous and physically meaningful metric for assessing the true binding potential of generated molecules. This limitation is evident in models such as Pocket2mol and VoxBind, which show substantial performance degradation upon docking. Other benchmarks exhibit distinct weaknesses; for instance, D3FG's fragment-based prior does not ensure strong binding energies, the FLAG model compromises interaction quality for energy optimization, and both LiGAN and DiffBP are constrained by low chemical validity. In contrast, our proposed IBEX model demonstrates superior and consistent performance, achieving state-of-the-art efficacy while maintaining top-tier chemical properties. While its initial Vina Score is not the highest, its physically more realistic Vina Dock score of -8.09 is exceptional. This is further substantiated by a significant improvement in the docking success rate (IMP), which increased from a baseline of 53% with TargetDiff to 64%. This suggests that molecules generated by IBEX assimilate knowledge of the overall protein structure, enabling strong interactions with the target. Ablation experiments confirm that the scaffold hopping is crucial for these performance improvements, while the side-chain decoration contributes minimally. Furthermore, a comparison on the CBGBench, illustrated in Figure 7, confirms that IBEX generates molecules with exceptionally balanced and high-performing docking characteristics relative to other generative models.

**IBEX Balances Strong Binding with Practical Feasibility.** Figure 8 presents docking poses and two-dimensional structures for four receptors. IBEX shows the lowest Vina score in every pocket. These energies correlate with tighter placement inside the catalytic cavity. Ligands generated by De-compDiff and MolCRAFT either extend beyond the binding pocket or leave the hydrophobic clefts

unfilled. IBEX orients polar atoms toward canonical hydrogen-bond donors or acceptors. Aromatic scaffolds sit flush with hydrophobic shelves. This geometry preserves high drug-likeness and modest synthetic cost. DecompDiff can reach low energies but its molecules carry long flexible chains that lower QED and raise SA. MolCRAFT maintains a cleaner chemical profile, yet it often leaves void space, which weakens binding. TargetDiff shows the weakest complementarity and acts only as an architectural control. IBEX and TargetDiff share the same network and sampling schedule. The only change is that IBEX is trained with scaffold-hopping pairs under an information-bottleneck objective. The observed gains therefore, stem from the training scheme rather than from model size or inference heuristics. These findings indicate that pocket-aware context steers generative diffusion toward chemically sensible and potent binders.

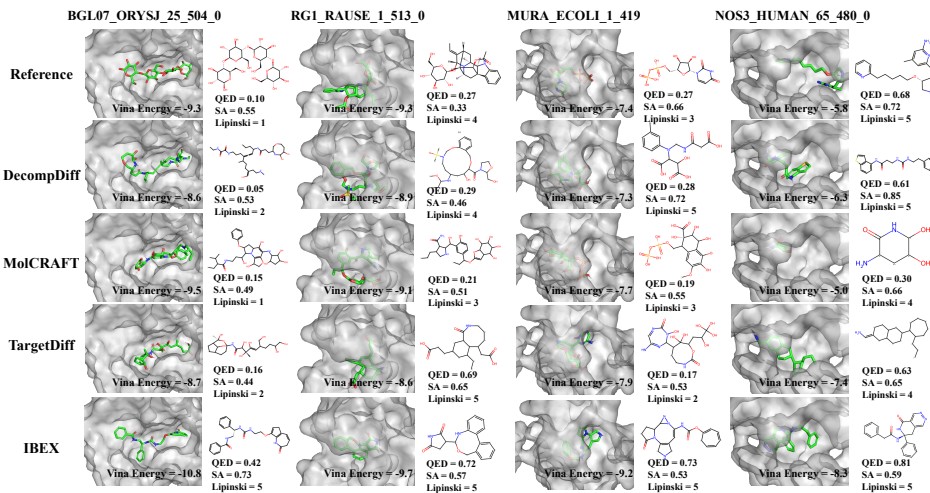

Figure 8: Docking poses and drug-likeness of median-score candidates on four CBGBench test pockets. Rows list the crystal reference and the molecule from DecompDiff, MolCRAFT, Target-Diff, and IBEX. Each ligand is depicted with its predicted pose (green sticks) and associated Vina energy, alongside its planar formula annotated with QED, SA, and Lipinski compliance.

**Batch Generative Performance Evaluation.** In Table 2 the metrics reported were obtained by generating 100 molecules for each of the 100 test pockets; whenever fewer than 100 structures were produced, the denominator was still fixed at 100. *Validity* denotes the fraction of chemically valid molecules, *Unique* counts the number of non-duplicate valid molecules, *Tanimoto* reports the mean pairwise fingerprint similarity among all generated molecules, and *Similar* is the mean similarity between each generated molecule and the reference ligand of its pocket. LiGAN exhibits very low validity and diversity, often producing identical or nearly identical

Table 2: Comparison of Diversity.

| Methods | Validity | Unique | Tanimoto | Similar |
|---|---|---|---|---|
| LiGAN | 0.42 | 0.3757 | 0.3249 | 0.3459 |
| POCKET2MOL | 0.75 | 0.7145 | 0.1181 | 0.0702 |
| D3FG | 0.77 | 0.7844 | 0.0926 | 0.0825 |
| DECOMPDIFF | 0.89 | 0.8429 | 0.1394 | 0.1469 |
| TARGETDIFF | 0.96 | 0.9524 | 0.1063 | 0.0976 |
| MOLCRAFT | 0.95 | 0.8828 | 0.1251 | 0.1154 |
| VOXBIND | 0.74 | 0.7418 | 0.1051 | 0.0998 |
| IBEX | 0.96 | 0.9507 | 0.1126 | 0.0761 |

molecules, and autoregressive baselines show the same limitation. To examine out-of-distribution performance, we further generated 2000 molecules for the previously unseen pocket 9F7W Useini et al. (2024) using Pocket2Mol and our IBEX model; after deduplication Pocket2Mol retained only 217 unique molecules, whereas IBEX preserved 1706, underscoring the superior diversity delivered by diffusion-based generators. Owing to its novel training regime, IBEX sustains state-of-the-art validity, uniqueness, and diversity even in this zero-shot *de novo* setting.

## 5 CONCLUSION

We introduce IBEX, an information-bottleneck-explored coarse-to-fine pipeline, and demonstrate both theoretically and experimentally its feasibility in extracting maximal information from extremely scarce datasets. This work establishes a theoretical and practical foundation for future structure-based drug design paradigms by seamlessly integrating information theory with physics-based optimization.

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
