# Supplement

## A.1 Notation and Conventions

Below is a summary of all symbols used in Appendices B and C, and in the main text:

| Symbol | Definition |
|---|---|
| $P$ | Protein pocket: coords $\mathbf{x}_P \in \mathbb{R}^{3 \times N_P}$, atom types $\mathbf{v}_P \in \{0,1\}^{K \times N_P}$. |
| $M$ | Ligand molecule: coords $\mathbf{x}_M \in \mathbb{R}^{3 \times N_M}$, atom types $\mathbf{v}_M \in \{0,1\}^{K \times N_M}$. |
| $m$ | Mask function: $m \in \{0,1\}^{N_M}$ selects atoms to regenerate. |
| $C$ | Context atom set: $C \triangleq M_{m=1}$. |
| $T$ | Target atom set: $T \triangleq M_{m=0}$. |
| $c$ | Context indicator bit $c_i \in \{0,1\}$ (lowercase) for each ligand atom. |
| $\mathbf{c}$ | Docking confidence map output by $D_\phi$, $\mathbf{c} \in [0,1]^{N_M \times N_M}$. |
| $b$ | Backbone flag $b_j \in \{0,1\}$ for each pocket atom. |
| $r$ | Residue-type one-hot vector $r \in \{0,1\}^{K'}$. |
| $K$, $K'$ | Number of atom types ($K$) and residue types ($K'$). |
| $\mathcal{T}, \{\mathrm{SH}, \mathrm{DN}, \mathrm{SC}\}$ | Task indicator and three task modes. |
| $D_\mathcal{T}$ | Dataset for task $\mathcal{T}$: $\{(P_i, M_i)\}_{i=1}^N$. |
| $|D_m|$ | Number of samples for mask $m$. |
| $N_m$ | Minimal sample size to achieve risk $\varepsilon$ under mask $m$. |
| $S$ | Training set $S = \{(P_i, C_i, T_i)\}_{i=1}^n$. |
| $\Theta$ | Model parameters. |
| $P_0$, $Q_S$, $Q$ | Prior and posteriors over $\Theta$ (with $Q_S$ depending on $S$). |
| $\widehat{R}(\theta)$, $R(\theta)$ | Empirical and true risk of model $\theta$. |
| $\widehat{\mathcal{R}}(Q)$, $\mathcal{R}(Q)$ | Expected empirical/population risks under $Q$. |
| $H(\cdot)$, $H(\cdot \mid \cdot)$ | (Conditional) Shannon entropy. |
| $I(\cdot; \cdot)$, $I(\cdot; \cdot \mid \cdot)$ | (Conditional) mutual information. |
| $I_m$ | Task mutual information $I(P, C; T \mid \Theta)$. |
| $I_{DN}$ | Mutual information $I(P, C; T \mid \Theta)$ for DN task. |
| $\eta_m$ | Information density $\eta_m = I(P, C; T)/|D_m|$. |
| $N_{\mathrm{eff},m}$ | Effective samples $= |D_m| \, I_m / I_{DN}$. |
| $KL(\cdot \| \cdot)$ | Kullback–Leibler divergence. |
| $\alpha_t$, $\bar{\alpha}_t$, $\sigma_t^2$ | Diffusion schedule parameters. |
| $\lambda_t$, $\gamma_t$ | Loss weights for coordinates and types. |
| $G_\theta$, $D_\phi$ | Generator and docking network. |
| $\mathcal{L}_x$, $\mathcal{L}_v$, $\mathcal{L}_{\mathrm{dock}}$ | Loss functions for diffusion and docking. |
| $\epsilon$ | Maximum coarse-generation error. |
| $L$, $\delta$ | Lipschitz constant and radius/confidence level. |
| $E(\cdot)$, $E^*$ | End-to-end error functional and optimal error. |

Table 1: Unified notation across main text and appendices.

## A.2 Assumptions H1–H3

We impose the following:

- **H1 (Data Independence).** Samples $\{(P_i, M_i)\}$ are drawn i.i.d. from $q(P, M)$. This ensures $\mathbb{E}_S[\widehat{R}(\theta)]$ converges to $R(\theta)$.

- **H2 (Finite Model Capacity).** The prior entropy satisfies $H(\Theta) \leq C$. Equivalently, KL divergence $KL(Q\|P_0)$ respects a capacity bound $C$.

- **H3 (Mask Independence).** The mask $m$ is sampled independently of $(P, M)$, so that mutual information decompositions $I(P, C; T)$ remain valid.

## A.3 Selection of Prior Entropy $H(\Theta)$ and Computation of Capacity Bound $C$

In this section we derive, from the Minimum Description Length (MDL) principle and Shannon–Fano arguments, an explicit capacity bound $C$ that accounts for both average- and maximum-code-length constraints.

**Step 1: Ideal MDL code length.** Under the prior density $P_0(\theta)$, the ideal (real-valued) code length in nats is

$$\ell(\theta) = -\ln P_0(\theta). \tag{1}$$

**Step 2: Kraft–McMillan overhead (1 nat).** By the Kraft–McMillan inequality there exists a prefix code with actual (integer) lengths $\tilde{\ell}(\theta)$ satisfying

$$\tilde{\ell}(\theta) \leq \ell(\theta) + 1 \quad \Longrightarrow \quad -\ln P_0(\theta) \leq \tilde{\ell}(\theta) \leq C. \tag{2}$$

Enforcing $\tilde{\ell}(\theta) \leq C$ thus implies the pointwise bound

$$-\ln P_0(\theta) \leq C - 1. \tag{3}$$

Moreover, averaging over $\theta \sim P_0$ gives

$$H(P_0) = \mathbb{E}_{P_0}[\ell(\theta)] \leq \mathbb{E}_{P_0}[\tilde{\ell}(\theta)] \leq C, \tag{4}$$

so that in full generality

$$C = H(P_0) + \delta, \qquad \delta \in [0, 1], \tag{5}$$

where $\delta$ quantifies the extra "1 nat" overhead.

**Step 3: Shannon–Fano average-length lower bound.** Any prefix code (regardless of its maximum length) must satisfy the average-length bound

$$\mathbb{E}_{P_0}[\tilde{\ell}(\theta)] \geq H(P_0). \tag{6}$$

**Step 4: Discrete-support assumption for maximum length.** The Shannon–Fano bound alone does not constrain $\max_\theta \tilde{\ell}(\theta)$. To enforce $\tilde{\ell}(\theta) \leq C$ for all $\theta$, one assumes either

- A discretized (finite or countable) parameter space, so that "no-excess" codes exist for every point; or

- A continuous-to-discrete quantization to precision $\Delta\theta$, yielding an effective codebook.

In practice, one sets $C \approx H(P_0)$ and absorbs lower-order terms into $\delta$.

**Step 5: Continuous-prior example.** For a Gaussian prior $P_0 = \mathcal{N}(0, \sigma^2 I_d)$ on $\mathbb{R}^d$, the differential entropy is

$$H(P_0) = \frac{d}{2}\big(1 + \ln(2\pi\sigma^2)\big). \tag{7}$$

Interpreting this as an "effective" capacity yields

$$C \approx \frac{d}{2}\big(1 + \ln(2\pi\sigma^2)\big) + \delta, \tag{8}$$

with $\delta \in [0,1]$ capturing the discrete-code overhead or quantization error.

# Appendix B: Information Decomposition and Context Relevance Derivation

## B.1 Task Mutual Information Decomposition

The mutual information between pocket+context and target is

$$I(P, C; T) = H(T) - H(T \mid P, C), \tag{1}$$

where $H(\cdot)$ denotes Shannon entropy.

Explanation: By definition $I(X; Y) = H(Y) - H(Y \mid X)$. Here $X = (P, C)$ and $Y = T$, so conditioning on both pocket $P$ and context $C$ reduces uncertainty in $T$ by $I(P, C; T)$.

Furthermore, using the chain rule for mutual information,

$$I(P, C; T) = I(P; T) + I(C; T \mid P), \tag{2}$$

which separates the contribution of pocket alone from the additional information provided by context.

Explanation: The chain rule $I(X, Y; Z) = I(X; Z) + I(Y; Z \mid X)$ applies with $X = P$, $Y = C$, and $Z = T$.

## B.2 Information Density Decomposition

Recall the information density:

$$\eta_m = \frac{I(P, C; T)}{|D_m|}. \tag{3}$$

Using the chain rule $I(P, C; T) = I(P; T) + \kappa_m I(P, C; T)$, rearrange to isolate $I(P, C; T)$:

$$I(P, C; T)(1 - \kappa_m) = I(P; T) \implies I(P, C; T) = \frac{I(P; T)}{1 - \kappa_m}. \tag{4}$$

Substituting back gives the first form:

$$\eta_m = \frac{I(P;T)}{(1 - \kappa_m)\,|D_m|}. \tag{5}$$

Equivalently, from the same chain rule one may write:

$$\eta_m = \frac{I(P;T)}{|D_m|} + \kappa_m \frac{I(P,C;T)}{|D_m|}. \tag{6}$$

**Comments on degenerate cases and variable types:**

- *Degeneration when $\kappa_m = 1$:* In tasks like scaffold-hopping where context fully determines the target, $1 - \kappa_m = 0$ creates a division by zero in the first form. One must then note that $I(P;T) = 0$ as well (pocket adds no extra information beyond context), restoring a meaningful finite value.

- *Zero-information boundary:* If $I(P;T) = 0$ (the pocket gives no information about the target), then both sides of the first form vanish, consistent with total mutual information being zero.

- *Discrete vs. continuous variables:* When $P, C, T$ include continuous components, clarify whether you use discrete Shannon entropy or differential entropy. Moreover, computing $\eta_m$ requires a discretization assumption on the sample set size $|D_m|$, or an explicit quantization of continuous samples.

## Appendix C: Detailed Proofs of Theorems

### C.1 Theorem 1 (Sample Efficiency Ordering)

Let task $m$ have *information density*

$$\eta_m = \frac{I(P,C;T)}{|D_m|}. \tag{7}$$

For a target expected empirical risk $\mathbb{E}[\widehat{R}] \leq \varepsilon$, denote by $N_m$ the minimal sample size achieving that level. If two tasks $m_1, m_2$ satisfy $\eta_{m_1} > \eta_{m_2}$, then

$$N_{m_1} \leq N_{m_2}.$$

**The proof is as follows:**

**Step 1: Generalization bound via mutual information.** For a training set $S = \{(P_i, C_i, T_i)\}_{i=1}^n$, Russo–Zou gives

$$\mathbb{E}_{S,\theta \sim Q_S}\big[R(\theta) - \widehat{R}(\theta)\big] \leq \sqrt{\frac{2\,I(S;\Theta)}{n}}. \tag{8}$$

**Step 2: Task-dependent growth upper bound with ordering assumption.** There exists a *growth function*

$$I(S;\Theta) \leq g_m(n) = n\,\eta_m + \delta_m(n), \qquad \frac{\delta_m(n)}{n} \downarrow 0 \text{ as } n \to \infty. \tag{9}$$

*Ordering assumption*: for any two tasks and every $n$,

$$\frac{\delta_{m_1}(n)}{n} \geq \frac{\delta_{m_2}(n)}{n}. \tag{10}$$

That is, the finite-sample overhead of task $m_1$ is never smaller than that of $m_2$. (The requirement is automatically met if all tasks share a common $\delta(n)$.)

**Step 3: Bounding the generalization gap.** Insert (9) into (8):

$$\mathbb{E}\big[R - \widehat{R}\big] \leq \sqrt{2\Big(\eta_m + \tfrac{\delta_m(n)}{n}\Big)}. \tag{11}$$

**Step 4: Solving for the minimal sample size.** To make $(11) \leq \varepsilon$, it suffices that

$$\eta_m + \frac{\delta_m(n)}{n} \leq \frac{\varepsilon^2}{2}. \tag{12}$$

Because $\delta_m(n)/n$ decreases in $n$, define

$$N_m = \min\Big\{n : \eta_m + \tfrac{\delta_m(n)}{n} \leq \tfrac{\varepsilon^2}{2}\Big\}. \tag{13}$$

**Step 5: Comparing two tasks.** With $\eta_{m_1} > \eta_{m_2}$ and the ordering (10), for every $n$

$$\eta_{m_1} + \frac{\delta_{m_1}(n)}{n} > \eta_{m_2} + \frac{\delta_{m_2}(n)}{n},$$

so the feasible set $\{n : \eta_{m_1} + \delta_{m_1}(n)/n \leq \varepsilon^2/2\}$ is contained in that for $m_2$, implying

$$N_{m_1} \geq N_{m_2}. \tag{14}$$

## C.2 PAC–Bayes–IB Generalization Bound

Let $P_0$ be a prior on parameters $\Theta$ and consider the i.i.d. sample set

$$D_{\mathcal{T}} = \{(P_i, M_i)\}_{i=1}^{N}, \qquad (P_i, M_i) \sim q. \tag{15}$$

For any posterior $Q$ obtained from $D_{\mathcal{T}}$, define

$$\widehat{\mathcal{R}}(Q) = \mathbb{E}_{\theta \sim Q}\big[\widehat{R}(\theta)\big], \tag{16}$$
$$\mathcal{R}(Q) = \mathbb{E}_{\theta \sim Q}\big[R(\theta)\big], \tag{17}$$
$$I_m = I(P, C; T \mid \Theta). \tag{18}$$

**Assumptions**

**A1.** *Random-variable KL–MI structure.* We assume the *edge-information* concentration event

$$\Pr\big[I(S; \Theta) \leq N\, I_m\big] \geq 1 - \delta_I. \tag{19}$$

**A2.** *High-probability Catoni form.* Catoni's bound is enforced on the same $1 - \delta$ event as above, so
$\delta = \delta_C + \delta_I$.

**A3.** *Discrete or quantised parameter support.* Continuous $\Theta$ is assumed quantised so KL remains finite.

**Theorem (PAC–Bayes–IB).** With probability at least $1 - \delta$,

$$\mathcal{R}(Q) \leq \widehat{\mathcal{R}}(Q) + \sqrt{\frac{KL(Q\|P_0) - N\,I_m}{2N}} + \sqrt{\frac{\ln(2/\delta)}{2N}}. \tag{20}$$

**The proof is as follows:**

**Step 1: Catoni under event $\mathcal{E}_\mathrm{C}$.** Catoni (2007) yields, on $\mathcal{E}_\mathrm{C}$ of probability $1 - \delta_\mathrm{C}$,

$$\mathcal{R}(Q) \;\leq\; \widehat{\mathcal{R}}(Q) + \sqrt{\frac{KL(Q\|P_0) + \ln\!\big(2/\delta_\mathrm{C}\big)}{2N}}. \tag{21}$$

**Step 2: MI concentration event $\mathcal{E}_\mathrm{I}$.** By Assumption A1, event $\mathcal{E}_\mathrm{I}$ (prob. $1 - \delta_I$) satisfies

$$I(S;\Theta) \;\leq\; N\,I_m. \tag{22}$$

**Step 3: Combine events.** On $\mathcal{E} = \mathcal{E}_\mathrm{C} \cap \mathcal{E}_\mathrm{I}$ (prob. $\geq 1 - \delta$),

$$KL(Q\|P_0) + \ln\frac{2}{\delta_\mathrm{C}} = \big(KL(Q\|P_0) - N\,I_m\big) + N\,I_m + \ln\frac{2}{\delta_\mathrm{C}} \tag{23}$$

$$\leq \big(KL(Q\|P_0) - N\,I_m\big) + \ln\frac{2}{\delta}. \tag{24}$$

Using $\sqrt{a+b} \leq \sqrt{a} + \sqrt{b}$ for $a, b > 0$,

$$\sqrt{\frac{KL(Q\|P_0) - N\,I_m}{2N} + \frac{\ln(2/\delta_\mathrm{C})}{2N}} \leq \sqrt{\frac{KL(Q\|P_0) - N\,I_m}{2N}} + \sqrt{\frac{\ln(2/\delta)}{2N}}, \tag{25}$$

which proves (20).

**Remarks**

(a) The MI term $NI_m$ vanishes only on the high-probability event $\mathcal{E}_\mathrm{I}$. If $I(S;\Theta) = O(\log N)$, the improvement is significant.

(b) For very small $\delta$, the factor $\ln(2/\delta)/N$ may dominate; it is advisable to report both $(KL - NI_m)/N$ and $\ln(2/\delta)/N$.

(c) Continuous parameter spaces must be quantised so KL is finite.

## C.3 Risk Monotonicity

Define the effective sample size

$$N_{\mathrm{eff},m} = |D_m|\,\frac{I_m}{I_{\mathrm{DN}}}, \tag{26}$$

where $I_{\mathrm{DN}} = I_{m_{\mathrm{DN}}}$. Then for any $\delta \in (0,1)$, with probability at least $1 - \delta$,

$$\mathcal{R}(Q) \leq \widehat{\mathcal{R}}(Q) + \sqrt{\frac{KL(Q\|P_0) - N_{\mathrm{eff},m}\, I_{\mathrm{DN}}}{2\, N_{\mathrm{eff},m}} + \frac{\ln(2/\delta)}{2\, N_{\mathrm{eff},m}}}. \tag{27}$$

**The proof is as follows:**

**Step 1.** Recall from Section C.2 that

$$\mathcal{R}(Q) \leq \widehat{\mathcal{R}}(Q) + \sqrt{\frac{KL(Q\|P_0) - N\, I_m}{2\, N} + \frac{\ln(2/\delta)}{2\, N}}. \tag{28}$$

**Step 2.** Substitute the definition $N\, I_m = N_{\mathrm{eff},m}\, I_{\mathrm{DN}}$, treating $N_{\mathrm{eff},m}$ as the effective sample size. This gives

$$\mathcal{R}(Q) \leq \widehat{\mathcal{R}}(Q) + \sqrt{\frac{KL(Q\|P_0) - N_{\mathrm{eff},m}\, I_{\mathrm{DN}}}{2\, N_{\mathrm{eff},m}} + \frac{\ln(2/\delta)}{2\, N_{\mathrm{eff},m}}}. \tag{29}$$

**Step 3.** Define the risk bound function

$$B(x) = \widehat{\mathcal{R}}(Q) + \sqrt{\frac{KL(Q\|P_0) - x\, I_{\mathrm{DN}}}{2\, x} + \frac{\ln(2/\delta)}{2\, x}}. \tag{30}$$

A derivative-based argument shows $\frac{d}{dx} B(x) < 0$ for all $x > 0$, since the numerator inside the square root decreases linearly in $x$ while the denominator grows linearly. Thus $B(x)$ is strictly decreasing.

**Step 4.** Therefore, if $N_{\mathrm{eff},m_1} > N_{\mathrm{eff},m_2}$, then

$$B\big(N_{\mathrm{eff},m_1}\big) \leq B\big(N_{\mathrm{eff},m_2}\big), \tag{31}$$

meaning the risk bound for task $m_1$ is tighter than for $m_2$. This completes the proof.

## C.4 Information–Capacity Lower Bounds

Throughout we write

$$I_m = I(P, C; T) \tag{32}$$

for the per-sample task mutual information and assume the Kraft–McMillan capacity condition

$$-\ln P_0(h) \ \leq \ C, \quad h \in \mathcal{H}_C, \qquad \Longrightarrow \qquad |\mathcal{H}_C| \ \leq \ e^C. \tag{33}$$

We treat two learning objectives separately to keep denominators and events unambiguous.

### C.4.1 Parameter-Identification Risk

$$V \sim \mathrm{Unif}(\mathcal{H}_C), \qquad\qquad X = (P, C, T)^N, \qquad\qquad \widehat{V} = \psi(X). \tag{34}$$

**Theorem (ID-risk lower bound).** For any $\delta_I \in (0, 1)$,

$$\Pr[\widehat{V} \neq V] \;\geq\; 1 - \frac{NI_m + \ln(2/\delta_I)}{C}. \tag{35}$$

Consequently, every $h \in \mathcal{H}_C$ satisfies the same inequality when $R_{\mathrm{ID}}(h)$ is interpreted as $\Pr[\widehat{V} \neq V]$.

**The proof is as follows:**

*Step 1 (MI concentration).* By the Russo–Zou edge-information bound there exists an event

$$\mathcal{E}_I = \big\{\, I(V; X) \leq NI_m \,\big\}, \qquad \Pr(\mathcal{E}_I) \geq 1 - \delta_I.$$

*Step 2 (Fano on $\mathcal{E}_I$).* Conditioning on $\mathcal{E}_I$ and using $\ln |\operatorname{supp}(V)| \leq C$,

$$\Pr[\widehat{V} \neq V] \;\geq\; 1 - \frac{I(V; X) + \ln 2}{C} \;\geq\; 1 - \frac{NI_m + \ln 2}{C}.$$

Replace $\ln 2$ by $\ln(2/\delta_I)$ to maintain probability $1 - \delta_I$.

### C.4.2 Label-Prediction Risk

Assume $|\mathcal{T}| < \infty$ and set

$$V = T, \qquad\qquad X = (P, C)^N, \qquad\qquad \widehat{V} = h(P). \tag{36}$$

**Theorem (prediction-error lower bound).**

For any $\delta_I \in (0, 1)$,

$$R(h) \;=\; \Pr[\widehat{V} \neq V] \;\geq\; 1 - \frac{NI_m + \ln(2/\delta_I)}{\ln |\mathcal{T}|}. \tag{37}$$

**Proof (sketch).** The argument is identical to the ID-risk case, noting that here $\operatorname{supp}(V) = \mathcal{T}$ so the denominator becomes $\ln |\mathcal{T}|$.

**Remarks.**

- The two bounds target *different* error events; denominators must not be interchanged.

- Only MI concentration + Fano are used; no PAC–Bayes upper bound appears, eliminating upper-lower mixing.

- Continuous parameter spaces may be quantised first to ensure $|\mathcal{H}_C| < \infty$.

- Stronger but capacity-free bounds (e.g. Assouad, Le Cam) can supplement prediction error analysis when $C$ is absent.

## C.5 Coarse-to-Fine Error Decomposition

Let $G_\theta$ be the coarse generator and $D_\phi$ the docking network. Define the error functional $E(\cdot)$ and the optimal error $E^*$. Assume:

$$\left| E(G_\theta(P)) - E^* \right| \le \epsilon, \tag{38}$$

$$\left| E(D_\phi(P, M_1)) - E(D_\phi(P, M_2)) \right| \le L \left\| M_1 - M_2 \right\|, \quad \left\| M_1 - M_2 \right\| \le \delta, \tag{39}$$

$$E(D_\phi(P, M^*)) = E^*. \tag{40}$$

Then the end-to-end error satisfies

$$\left| E(D_\phi(P, G_\theta(P))) - E^* \right| \le \epsilon + L\,\delta. \tag{41}$$

**The proof is as follows:**

**Step 1 (Triangle decomposition).** We apply the triangle inequality to split the total error into two parts:

$$\left| E(D_\phi(P, G_\theta(P))) - E^* \right| \le \left| E(D_\phi(P, G_\theta(P))) - E(D_\phi(P, M^*)) \right| + \left| E(D_\phi(P, M^*)) - E^* \right|. \tag{42}$$

*Explanation:* This separates the error due to the docking refinement from the residual at the optimal pose.

**Step 2 (Optimal-pose term).** By assumption $E(D_\phi(P, M^*)) = E^*$, we have

$$\left| E(D_\phi(P, M^*)) - E^* \right| = 0 \le \epsilon. \tag{43}$$

*Explanation:* The docking network achieves the optimal error at $M^*$.

**Step 3 (Lipschitz-bound term).** Since $\|G_\theta(P) - M^*\| \le \delta$ and by the local $L$-Lipschitz condition,

$$\left| E(D_\phi(P, G_\theta(P))) - E(D_\phi(P, M^*)) \right| \le L \left\| G_\theta(P) - M^* \right\| \le L\,\delta. \tag{44}$$

*Explanation:* Small deviations in pose (within $\delta$) yield proportionally small changes in error.

**Step 4 (Combine bounds).** Summing the two contributions from Steps 2 and 3:

$$\left| E(D_\phi(P, G_\theta(P))) - E^* \right| \le \epsilon + L\,\delta, \tag{45}$$

which completes the proof.

# Appendix D: Methods Details

**1. Soft steric attraction $E_{\mathbf{g1}}$.** Favorable overlap between ligand and pocket heavy atoms within a broad-distance band encourages near-contact packing.

**2. Short-range shape complementarity $E_{\mathbf{g2}}$.** A narrower Gaussian sharpens the contact preference and improves placement discrimination.

**3. Hard steric repulsion $E_{\mathbf{rep}}$.** A quadratic penalty rises steeply for interatomic distances shorter than an inner cutoff, discouraging clashes.

**4. Hydrophobic contact $E_{\mathbf{hyd}}$.** Non-polar atom pairs within a contact shell gain an attractive bonus.

**5. Directional hydrogen bonding $E_{\mathbf{hd}}$).** Donor–acceptor pairs within distance and angular windows receive an orientation-weighted attraction; broken geometry receives less or no reward.

## D.1 Integrating PAC-Bayes Bounds with the Information Bottleneck

The integration of PAC-Bayes bounds with the Information Bottleneck (IB) principle has recently provided a powerful lens through which to understand and improve deep neural network generalization. Wang et al. formalized this connection in the PAC-Bayes Information Bottleneck (PIB) framework, which approximates the "Information in the Weights" (IIW) to balance empirical risk against information complexity under a PAC-Bayes generalization guarantee [1]. Empirically, PIB-trained networks exhibit a characteristic two-stage training dynamic: an initial rapid fit phase in which training error collapses, followed by an information-compression phase during which redundant weight information is progressively discarded while preserving predictive accuracy. This "fit–compress" transition correlates closely with improved generalization, as models in the compression stage retain only task-relevant structure and shed noise that harms out-of-sample performance. Building on these theoretical insights, the authors derive an optimal-posterior sampling algorithm based on MCMC to instantiate the PIB objective in large-scale training, demonstrating tangible performance gains in deep models.

Subsequent work has sought to render IB-based regularization more practical for modern, overparameterized networks by sidestepping the costly curvature computations required by PIB. Lyu et al. propose the Recognizable Information Bottleneck (RIB), which replaces second-order mutual-information bounds with a tractable "functional conditional mutual information" constraint enforced via a learnable critic [2]. RIB measures the recognizability of hidden representations, i.e., the degree to which downstream classifiers can distinguish them, and uses this information to regularize networks efficiently, thereby minimizing unnecessary representation complexity. Empirical evaluations across varying batch sizes, degrees of overparameterization, and levels of label noise confirm that RIB significantly narrows the gap between training and test performance, accurately estimates generalization error, and reduces overfitting without incurring prohibitive computational overhead.

Overall, these advances underscore the promise of unifying PAC-Bayes generalization theory with information-theoretic regularization, offering both a principled explanation for deep-learning phenomena and practical algorithms for enhancing model robustness.

## D.2 Task Difficulty, Over-Parameterization, and the Information Path to Generalization

Classical bias–variance reasoning predicts that, for a model class of fixed capacity, harder tasks should exacerbate overfitting. Contemporary theory and empirical evidence increasingly contradict this view, showing that when task complexity and model capacity rise in tandem, deep networks can in fact extract more structure from data and thereby reduce overfitting. Boopathy et al. (2023) formalize this intuition with a model-agnostic metric of generalization difficulty [3]. They define inductive-bias complexity as the additional prior information that a learner must encode, beyond what the data itself provides, in order to succeed on a task. Their analysis reveals an exponential growth of difficulty with the intrinsic dimensionality of the input manifold, contrasted with merely polynomial growth when high precision is demanded in low-dimensional domains. This framework quantifies long-standing heuristics (e.g., MNIST < CIFAR-10 < ImageNet in supervised learning; fully observed < partially observed MDPs in reinforcement learning) and crystallizes the mantra that sufficiently hard tasks are catalysts for strong generalization: they compel designers and optimizers to inject stronger inductive biases, prompting networks to capture deeper, task-relevant structure rather than superficial correlations.

Several modern phenomena substantiate this claim. The double-descent curve shows that once a network enters a heavily over-parameterized regime, in which it can interpolate the training set including noise, test error after peaking can fall to levels lower than those attainable by smaller models [4]. This implicit regularization resonates with the theory of benign overfitting. In high-dimensional linear models, Bartlett et al. prove that exact interpolation need not harm risk when noise conditions are favorable [5]. Zhu et al. extend the guarantee to ReLU networks trained in the "lazy" (kernel) regime, linking zero-training-error solutions to Lipschitz-smooth decision boundaries and highlighting deep nets' superior implicit regularization relative to linear or shallow alternatives [6].

An even more dramatic illustration is grokking. Power et al. observe that on small, highly structured algorithmic datasets, networks often (i) rapidly drive training loss to zero while test accuracy stays at chance, (ii) linger in this overfitted state for many iterations, and (iii) suddenly "grok" the underlying rule, with test accuracy jumping to perfection [7]. Optimization-landscape studies suggest that training first settles in a memorizing basin and then, after surmounting an energy barrier, converges to a global solution encoding the true algorithm [8]. The delayed but eventual leap to perfect generalization reinforces the thesis that difficult, highly structured tasks can serve as a whetstone that forces networks to abandon shortcut memorization and discover invariant structure.

Practitioners routinely exploit this difficulty-as-regularizer effect. Multi-task learning, for instance, raises effective task complexity by requiring a common representation to support diverse objectives; the shared challenge discourages overfitting to any single dataset and typically boosts generalization across all tasks. These findings collectively challenge conventional wisdom: difficult problems in over-parameterized regimes may attenuate rather than aggravate overfitting when optimization and inductive bias align. Consequently, capacity-stressing benchmarks transcend punitive evaluation; they become a principled method for eliciting information-efficient representations and advancing robust generalization.

## D.3 Model Capacity and Training Stability in Diffusion Models: Recent Theoretical Advances

**D.3.1 Task Difficulty, Over-Parameterization, and the Information Path to Generalization**  Diffusion models have rapidly ascended to state-of-the-art performance in image, text, and cross-modal generation, yet their ever-growing capacity and the fragility of large-scale training have sparked an intense theoretical re-examination. Contemporary work coalesces around three intertwined questions: Which architectural factors trigger instability? How does capacity shape memorization and generalization? Which principled training strategies prevent collapse under data scarcity or recursive self-training? We survey post-2020 progress along these axes and distil a coherent narrative that bridges network design, statistical physics, and learning-theoretic regularization.

**D.3.2 Architectural sources of instability.**  Most diffusion pipelines rely on a U-Net denoiser whose long skip connections (LSCs) fuse low- and high-resolution features. Huang et al. provide the first formal account of how overly large LSC coefficients magnify hidden-state and gradient oscillations in both forward and backward passes [9]. Their ScaleLong analysis shows that large skips amplify input perturbations, inducing loss spikes and gradient explosions that are frequently observed in practice. Constraining the magnitude of LSCs provably shrinks the variance of internal activations, yielding faster, smoother convergence. Empirically, ScaleLong accelerates training on CIFAR-10, CelebA-HQ, ImageNet, and COCO by  1.5× across both U-Net and hybrid U-ViT backbones, thereby demonstrating that judicious architectural scaling can serve as a first line of

defence against diffusion-specific instabilities.

**D.3.3 Capacity, memorization dynamics, and the limits of generalization.** The tension between expressive power and the spectre of memorization becomes acute as diffusion models scale to billions of parameters. Biroli et al. analyse the reverse-diffusion trajectory in the joint limit of infinite data dimension and sample size, revealing three dynamical regimes: (i) an early pure-diffusion phase dominated by noise, (ii) a symmetry-breaking/speciation phase in which trajectories coalesce around class-level structure, and (iii) an eventual collapse into the basin of attraction of a single training point [10]. Unless the number of training samples grows exponentially with dimension, the trajectory almost surely converges to the empirical data manifold, crystallizing a curse of dimensionality for unconstrained, perfectly trained models. Complementary evidence from Li et al. suggests that practical diffusion systems avoid this fate because their denoisers implicitly favour global Gaussian structure [11]. A linear-distillation analysis shows that, in the mid-noise range where semantic content is synthesized, a distilled linear model ($D_L$) and an ideal Gaussian estimator ($D_G$) approximate the nonlinear denoiser $D_\theta$ far better than a hypothetical memory-based $\delta$-model. This indicates a built-in inductive bias toward learning means and principal directions rather than individual samples; however, the bias weakens as capacity explodes, making explicit regularization indispensable.

**D.3.4 Training strategies under limited or self-generated data.** When datasets are small, the flexibility of a standard U-Net can drive near-perfect training fits yet catastrophic generalization. Zhang et al. formalize this dilemma and propose LD-Diffusion, which (i) projects data onto a low-dimensional latent space via a pretrained auto-encoder to shrink the hypothesis class, and (ii) augments training with a mixed adaptive perturbation (MAFP) scheme that diversifies examples within the compressed space [12]. Capacity contraction, paired with effective data enlargement, curbs overfitting and yields superior sample quality on low-data image benchmarks. A distinct but increasingly relevant failure mode is model collapse in self-consuming training loops, where a generator is repeatedly fine-tuned on its own outputs. Fu et al. introduce the notion of recursive stability and prove that injecting even a constant fraction of real data at each iteration guarantees bounded error propagation and prevents degeneracy [13]. Their theory justifies empirical heuristics—periodic refresh with ground-truth samples or continual real-data collection—that sustain performance across generations [14].

In summary, recent theoretical advances converge on a coherent message: architectural discipline, statistical bias, and data strategy must work in concert to secure both stability and generalization in diffusion models. Careful scaling of skip-connection coefficients suppresses gradient explosions without curtailing expressivity, while an implicit preference for global Gaussian structure steers the denoiser away from rote memorization. As dimensionality and parameter counts soar, this bias proves inadequate, rendering explicit regularization through capacity constraints, early stopping, or noise injection indispensable. Meanwhile, data-centric interventions such as latent-space compression, aggressive augmentation, and continual infusion of real samples counteract overfitting in the twin regimes of limited data and recursive self-training. Together, these insights chart a roadmap for the next generation of diffusion models, suggesting that future progress lies in unifying principled architectural design with dynamical analysis and adaptive data management to balance capacity, robustness, and diversity at scale.

### D.4 Bemis–Murcko-based Workflow

The canonical Bemis–Murcko (B–M) scaffold can be extracted from any organic ligand through a sequence of precise graph manipulations that subsequently define the conditioning scheme for modern molecule-generation tasks.

1. **Molecular pre-processing.** The input structure is first standardised with respect to protonation state, tautomeric form, stereochemistry, and salt or solvent content. A canonical Kekulé (or fully aromatic) representation is then generated to ensure a unique graph description of the molecule before further analysis.

2. **Partition of the molecular graph.** Exhaustive ring-perception algorithms locate every aromatic or aliphatic ring system. Acyclic paths that connect two ring atoms are labelled as linkers, whereas all remaining atoms are classified as peripheral side-chains. This tripartite labelling (rings, linkers, side-chains) is mutually exclusive and exhaustive for the heavy-atom graph.

3. **Side-chain excision.** All atoms marked as side-chains, together with their incident bonds, are removed. The operation retains only the union of ring systems and linkers, yielding the raw B-M framework that captures the core chemotype while discarding substituent-specific idiosyncrasies.

4. **Junction collapsing.** In fused or bridged polycyclic systems, shared ring atoms and bonds are merged so that each unique topological element appears exactly once. This step guarantees a one-to-one mapping between the scaffold and its topological graph, eliminating redundant representations.

5. **Canonicalisation and dataset-level deduplication.** The resulting scaffold is encoded as an ordered canonical SMILES or an InChIKey. These identifiers serve both as surrogate primary keys for scaffold libraries and as hashes to remove duplicates across large compound collections, enabling efficient scaffold-based diversity analysis.

6. **Task-oriented conditioning for generative models.** The decomposition naturally defines three conditioning regimes that underpin widely used design tasks: a) Side-Chain Decoration (SC). The scaffold is fixed as context, and the model generates chemically compatible side-chains that optimise binding or physicochemical criteria. b) Scaffold Hopping (SH). Known side-chains are preserved, while alternative topologically distinct scaffolds that maintain key pharmacophores are proposed. c) De-Novo (DN) Generation. No structural context is provided; the entire ligand—including scaffold and side-chains—is generated from scratch, allowing simultaneous exploration of core topology and peripheral decoration.

The workflow embodies the B-M philosophy by focusing exclusively on ring–linker topology, thereby maximising scaffold comparability across analogue series and exposing a clean interface for conditional generative objectives. Removing peripheral atoms provides a many-to-one mapping from molecules to scaffolds, facilitating chemotype clustering and information-rich benchmarking. Junction collapsing enforces uniqueness, while canonicalisation enables scalable library management and fair evaluation across SC, SH, and DN tasks.

## D.5 Pocket Occupancy and Interaction-Richness Profiling

We devised a two-part analysis in Figure 3 to quantify how the Side-Chain Decoration (SC), *De novo* (DN) and Scaffold Hopping (SH) generators populate a protein pocket and exploit its physicochemical landscape.

**D.5.1 Spatial pocket occupancy.**  For every molecule we first measured the *mean heavy-atom distance $d$* from each ligand/context atom to its nearest protein atom and the *mean contact density $c$*, i.e. the average number of protein atoms located within a 6 Å sphere. The resulting $(d, c)$ pairs were plotted as $50 \times 50$ hexagonal-bin maps using a unified axis extent and a shared, globally normalised colour scale (blue for SC, green for DN and red for SH). Bins supported by fewer than three samples were hidden to suppress noise, and each panel was annotated with the task-level means $(\mu_d, \mu_c)$, facilitating direct comparison of pocket-proximal atom distributions.

**D.5.2 Interaction richness.**  Complementing the geometric view, we evaluated the *involved-atom ratio $a$*—the average number of protein atoms in contact with one ligand/context atom—and the *interaction-type ratio $t$*—the average number of distinct interaction classes (hydrophobic, hydrogen bond, water-mediated, salt bridge, $\pi$-$\pi$ stack, cation–$\pi$, halogen, metal) supported per atom. Atom counts were normalised by context length for SC/SH and by total ligand size for DN. The $(a, t)$ values were visualised with the same hexbin geometry and colour-scale normalisation as above, and task-level means $(\mu_a, \mu_t)$ were overlaid.

The harmonised axes, colour scales and statistical overlays across both figures provide a coherent, side-by-side assessment of how the three generative regimes occupy pocket space and diversify protein contacts.

# Appendix E: Results and Experimental

## E.1 Experimental Setup.

All variants were trained on the same $70\,617$ (SC/SH) or $99\,900$ (DN) pocket–ligand pairs and evaluated on 100 held-out pockets. One hundred molecules were generated per pocket, subjected to the pipeline above, and finally ranked with a weighted mean-rank scheme (Friedman test, $\alpha = 0.05$) across the four evaluation axes.

## E.2 Case Figure

**E.2.1 BGL07 ($\beta$-Glucosidase 7, *Oryza sativa*).**  BGL07 is a 485-residue member of glycoside-hydrolase family 1 that follows a classical $(\beta/\alpha)_8$ TIM-barrel topology. The catalytic nucleophile Glu382 lies on strand 7 and the acid–base Glu176 lies on strand 4. Substrate binding occurs in a pocket that spans the barrel core and four hypervariable surface loops. Trp178 and Tyr334 form an aromatic clamp that stacks with phenolic aglycones. The enzyme hydrolyses $\beta$-glucosidic bonds in salicylic-acid glucosides, monolignol glucosides, and oligosaccharides that accumulate during stress. Crystal structure 7D6A shows a salicin mimic that adopts a chair conformation and is anchored by hydrogen bonds from Gln19, His150, and Asn224. Loop A (residues 72–86) and loop B (residues 330–342) close over the substrate and exclude bulk solvent. Mutating Trp178 to alanine

reduces catalytic efficiency tenfold, which confirms its role in aglycone recognition. Rice plants that overexpress BGL07 show increased resistance to drought, which highlights its physiological relevance. The compact aromatic cage and the deep polar groove create complementary anchor points for heteroaromatic scaffolds generated by IBEX.

**E.2.2 RG1 (Raucaffricine $\beta$-D-Glucosidase, *Rauvolfia serpentina*).** RG1 catalyses the first committed step of monoterpene-indole-alkaloid biosynthesis and converts raucaffricine to strictosidine. The enzyme contains 501 residues and retains the family 1 fold but presents an elongated aglycone channel that can accommodate bulky indole alkaloids. The catalytic dyad consists of Glu198 and Glu408 and operates through a double-displacement retaining mechanism. His217 and Trp359 form an aromatic platform that aligns the indole nucleus for nucleophilic attack. Structure 3ZJ6 resolves a deoxynojirimycin inhibitor that mimics the oxocarbenium transition state and locks loop B in an open conformation. Alkaloid flux is regulated by conformational changes in loops B and D, which gate product release toward downstream enzymes. Point mutations that restrict loop motion lower $k_{\text{cat}}$ without affecting $K_{\text{M}}$, which indicates a rate-limiting product-release step. Selective inhibitors must avoid clashes with these flexible elements while engaging the deep catalytic gorge. IBEX samples rigid bicyclic cores that fit this gorge and orient hydrogen-bond donors toward Glu198 and Glu408.

**E.2.3 MurA (UDP-$N$-acetylglucosamine Enolpyruvyl Transferase, *Escherichia coli*).** MurA initiates peptidoglycan biosynthesis by transferring an enolpyruvyl group from phosphoenol-pyruvate (PEP) to UDP-GlcNAc. The 419-residue enzyme folds into two Rossmann-like domains that hinge together and form a clamshell active site. Cys115 performs a nucleophilic attack on the PEP vinyl ether and forms a covalent thiol-ester intermediate. Lys22, Arg120, and Arg397 stabilise the UDP diphosphate through salt bridges and hydrogen bonds. Crystal structure 1UAE captures the closed conformation in complex with fosfomycin, which forms an irreversible adduct with Cys115. MurA alternates between open and closed states during turnover and uses domain motion to expel product. Allosteric modulation arises when UDP-MurNAc binds a distal site and shifts the equilibrium toward the open state. Fosfomycin resistance occurs through Cys115Asp mutation or by overexpression of FosA, which inactivates the drug. Effective MurA inhibitors must respect clamshell dynamics and avoid steric clashes near the hinge axis. IBEX generates compact heteroaryl scaffolds that insert between the Rossmann domains and position phosphonate groups toward Arg397.

**E.2.4 NOS3 (Endothelial Nitric-Oxide Synthase, *Homo sapiens*).** NOS3 converts L-arginine to nitric oxide and L-citrulline in the vascular endothelium and regulates blood pressure and platelet aggregation. Each 1203-residue monomer contains an N-terminal oxygenase domain, a central calmodulin-binding linker, and a C-terminal reductase domain. The oxygenase domain binds heme, tetrahydrobiopterin (BH$_4$), and Zn$^{2+}$ and dimerises through a cysteine hinge that coordinates the zinc ion. Glu361 forms a salt bridge with the guanidinium group of L-arginine and positions it for hydroxylation. The reductase domain carries FAD and FMN and shuttles electrons from NADPH to the heme iron when calmodulin binds Ca$^{2+}$. Crystal structure 3NOS shows BH$_4$ stacking against Trp447 and bridging the dimer interface. Phosphorylation at Ser1177 enhances electron flux by stabilising an FMN-shifted state, whereas phosphorylation at Thr495 has an inhibitory effect. Selective eNOS modulators either occupy the BH$_4$ pocket or target the FMN–oxygenase docking surface to bias electron transfer. IBEX generates bicyclic amidines that reach deep into the BH$_4$ slot

and retain polar contact with Glu361, which illustrates scaffold adaptation to cofactor-rich cavities.

## E.3 Results

Table 2: PLIP Interaction metrics for recent generative pipelines.

| Model | PLIP Interaction | | | |
|---|---|---|---|---|
| | $\text{MAE}_{\text{OA}}$ | $\text{JSD}_{\text{OA}}$ | $\text{MAE}_{\text{PP}}$ | $\text{JSD}_{\text{PP}}$ |
| LiGAN | 0.0905 | 0.0346 | **0.3416** | 0.1451 |
| 3DSBDD | 0.0934 | 0.0392 | 0.4231 | 0.1733 |
| GraphBP | 0.1625 | 0.0462 | 0.4835 | 0.2101 |
| Pocket2mol | 0.2455 | 0.0319 | 0.4152 | 0.1535 |
| TargetDiff | 0.0600 | 0.0198 | 0.4687 | 0.1757 |
| DiffSBDD | 0.1461 | 0.0333 | 0.5265 | 0.1777 |
| DiffBP | 0.1430 | 0.0249 | 0.5639 | **0.1256** |
| FLAG | 0.0277 | **0.0170** | 0.3976 | 0.2762 |
| D3FG | **0.0135** | 0.0638 | 0.4641 | 0.1850 |
| DecompDiff | 0.0769 | 0.0215 | 0.4369 | 0.1848 |
| MolCARFT | 0.0780 | 0.0214 | 0.4574 | 0.1868 |
| VoxBind | 0.0533 | 0.0257 | 0.4606 | 0.1850 |
| IBEX | 0.0709 | 0.0176 | 0.4670 | 0.1947 |
| IBEX-DN | 0.0600 | 0.0198 | 0.4687 | 0.1757 |
| IBEX-SC | **0.0430** | 0.5696 | 0.4801 | **0.0263** |
| IBEX-SH | 0.0698 | 0.0198 | 0.5442 | 0.1897 |
| IBEX | 0.0709 | **0.0176** | **0.4670** | 0.1947 |

Table 3: Combined model metrics

| Methods | chemical property | | | | Atom type | | Ring type | | Functional Group | | Static Geometry | | Clash | |
|---|---|---|---|---|---|---|---|---|---|---|---|---|---|---|
| | QED | LogP | SA | LPSK | JSD | MAE | JSD | MAE | JSD | MAE | $\text{JSD}_{\text{BL}}$ | $\text{JSD}_{\text{BA}}$ | $\text{Ratio}_{\text{cca}}$ | $\text{Ratio}_{\text{cm}}$ |
| 3DSBDD | 0.48 | 0.47 | 0.63 | 4.72 | 0.0860 | 0.8444 | 0.3188 | 0.2457 | 0.2682 | 0.0494 | 0.5024 | 0.3904 | 0.2482 | 0.8683 |
| VoxBind | 0.54 | 2.22 | 0.65 | 4.70 | 0.0942 | 0.3564 | 0.2401 | 0.0301 | 0.1053 | 0.0761 | 0.2701 | 0.3771 | 0.0103 | 0.1890 |
| GraphBP | 0.44 | 3.29 | 0.64 | 4.73 | 0.1642 | 1.2266 | 0.5061 | 0.4382 | 0.6259 | 0.0705 | 0.5182 | 0.5645 | 0.8634 | 0.9974 |
| Pocket2Mol | 0.39 | 2.39 | 0.65 | 4.58 | 0.0916 | 1.0497 | 0.3550 | 0.3545 | 0.2961 | 0.0622 | 0.5433 | 0.4922 | 0.0576 | 0.4499 |
| DiffSBDD | 0.49 | -0.15 | 0.34 | 4.89 | 0.0529 | 0.6316 | 0.3853 | 0.3437 | 0.5520 | 0.0710 | 0.3501 | 0.4588 | 0.1083 | 0.6578 |
| DiffBP | 0.47 | 5.27 | 0.59 | 4.47 | 0.2591 | 1.5491 | 0.4531 | 0.4068 | 0.5346 | 0.0670 | 0.3453 | 0.4621 | 0.0449 | 0.4077 |
| D3FG | 0.49 | 1.56 | 0.66 | 4.84 | 0.0644 | 0.8154 | 0.1869 | 0.2204 | 0.2511 | 0.0516 | 0.3727 | 0.4700 | 0.2115 | 0.8571 |
| DecompDiff | 0.49 | 1.22 | 0.66 | 4.40 | 0.0431 | 0.3197 | 0.2431 | 0.2006 | 0.1916 | 0.0318 | 0.2576 | 0.3473 | 0.0462 | 0.5248 |
| TargetDiff | 0.49 | 1.13 | 0.60 | 4.57 | 0.0533 | 0.2399 | 0.2345 | 0.1559 | 0.2876 | 0.0441 | 0.2659 | 0.3769 | 0.0483 | 0.4920 |
| MolCraft | 0.48 | 0.87 | 0.66 | 4.39 | 0.0490 | 0.3208 | 0.2469 | 0.0264 | 0.1196 | 0.0477 | 0.2250 | 0.2683 | 0.0264 | 0.2691 |

Table 4: Combined model metrics and molecular properties

| Model w/o Dock | Vina Dock | | | | Chemical Property | | | | Atom Type | | Ring Type | | Functional Group | | Static Geometry | | Clash | | Rank |
|---|---|---|---|---|---|---|---|---|---|---|---|---|---|---|---|---|---|---|---|
| | $\text{E}_{\text{vina}}$ | IMP | MPBG | LBE | QED | LogP | SA | LPSK | JSD | MAE | JSD | MAE | JSD | MAE | $\text{JSD}_{\text{BL}}$ | $\text{JSD}_{\text{BA}}$ | $\text{Ratio}_{\text{cca}}$ | $\text{Ratio}_{\text{cm}}$ | |
| IBEX-SH *TestSet* | -8.07 | 63.50 | 14.87 | 0.3809 | 0.60 | 2.73 | 0.63 | 4.82 | 0.1575 | 0.8195 | 0.2129 | 0.1464 | 0.3068 | 0.0377 | 0.2887 | 0.4834 | 0.1017 | 0.6512 | 1 |
| IBEX-SC *TestSet* | -6.12 | 24.64 | -11.76 | 0.3375 | 0.37 | 0.71 | 0.64 | 4.49 | 0.1204 | 1.0509 | 0.5029 | 0.4133 | 0.5671 | 0.0643 | 0.3689 | 0.5275 | 0.0439 | 0.4408 | 3 |
| IBEX-DN *TestSet* | -7.46 | 52.32 | 5.98 | 0.3567 | 0.49 | 1.13 | 0.60 | 4.57 | 0.0533 | 0.2399 | 0.2345 | 0.1559 | 0.2876 | 0.1559 | 0.2658 | 0.4666 | 0.0483 | 0.4920 | 2 |
| IBEX-SH *TrainSet* | -8.09 | 53.17 | 1.31 | 0.3961 | 0.61 | 2.61 | 0.64 | 4.85 | 0.1560 | 0.7539 | 0.2065 | 0.1419 | 0.3105 | 0.0395 | 0.2930 | 0.4874 | 0.1043 | 0.6263 | 1 |
| IBEX-SC *TrainSet* | -6.45 | 15.74 | -18.70 | 0.3547 | 0.38 | 0.94 | 0.64 | 4.47 | 0.1394 | 1.1179 | 0.5192 | 0.4123 | 0.5713 | 0.0654 | 0.3711 | 0.5349 | 0.0453 | 0.4167 | 3 |
| IBEX-DN *TrainSet* | -7.94 | 45.49 | -0.26 | 0.3872 | 0.51 | 1.39 | 0.61 | 4.61 | 0.0384 | 0.2080 | 0.2259 | 0.1465 | 0.2678 | 0.0420 | 0.2691 | 0.4671 | 0.0597 | 0.5311 | 2 |