# OpenReview forum: "IBEX: Information-Bottleneck-EXplored Coarse-to-Fine Molecular Generation under Limited Data"
_ICLR.cc/2026/Conference — ICLR 2026 Conference Withdrawn Submission_

### Official Review · Reviewer_fMUa · 2025-10-28

**Soundness:** 2
**Presentation:** 2
**Contribution:** 2
**Rating:** 2
**Confidence:** 5

**Summary:**

Decoupled, coarse-to-fine generator that learns to generate ligands with a diffusion model, then applies a lightweight physics refinement to polish poses. Trains only on the information-dense scaffold hopping task, and then zero-shot transfers that generator to full de-novo sampling at inference; the physics module is kept separate.

**Strengths:**

1. Shows SH is the most information-dense task and has the highest means across interaction/geometry axes and the broadest distributions (i.e., more informative training signals) with PAC-Bayes information-bottleneck view.SH exposes richer, more discriminative signal (higher information density) than DN/SC, tightening generalization bounds. Training on SH therefore structures the latent space so it transfers better when fully masking at test time (DN).

**Weaknesses:**

1. lacks reference papers related to BM-scaffold hopping generative models, recent de-novo generative models as well as performance comparison between

DiffHopp: A Graph Diffusion Model for Novel Drug Design via Scaffold Hopping (ICML 2023 Workshop)
TurboHopp: Accelerated Molecule Scaffold Hopping with Consistency Models (Neurips 2024)

Why not implement these models above for SH stage?

2. Figures are hard to understand. Typos(MOLCARFT -> MOLCRAFT) and the paper citation formats need improvement. The paper doesn't seem ready and clarity is low.

3. Questions below include weaknesses. Please refer to the questions.

**Questions:**

1. Vina score seems poor compared to other models. Why is this happening? Can you give results using posebuster?

2. can you provide the geometric metrics? (bond-length/angle distribution etc.) compared to other models.

3.  Why use TargetDiff? there are more recent well-performing models. also Targetdiff does not support bond diffusion. Can this method be extended to other generative models existing?

4. Physics-guided rigid-body pose refinement does not seem appealing. Pose refinement as well as dictionary based bond matching has been overcome with recent models.  extra post-hoc physics refinement seems like an unnecessary addition.

5. Please include FLOWR as well as other recent SOTA generative models. Also provide results on the SPINDR dataset which address existing data quality issues compared to crossdocked.

PoseBusters: AI-based docking methods fail to generate physically valid poses or generalise to novel sequences
FLOWR: Flow Matching for Structure-Aware De Novo, Interaction- and Fragment-Based Ligand Generation

---

### Official Review · Reviewer_7kcs · 2025-10-31

**Soundness:** 3
**Presentation:** 3
**Contribution:** 2
**Rating:** 4
**Confidence:** 4

**Summary:**

This paper addresses the problem of molecular generation given a protein pocket. First, the authors conduct a theoretical analysis of three problem settings commonly tackled in the field: de novo generation, scaffold hopping, and side chain decoration. The authors find that existing datasets best capture useful information about scaffold hopping, leading to the most information-dense data for the model to train on. Then, the authors develop a generative model, IBEX, based on this finding to do de novo generation, decoupling the training and inference process. The results show that the IBEX model generates molecules with better docking scores and other properties compared to baselines.

**Strengths:**

1. The theoretical analysis of the difficulties of each problem setting is a very interesting approach, and helps give some much-needed direction to the field of pocket-conditioned molecular generation. I think the metrics that the authors use make sense given the setup, especially looking at the gradient-SNR.
2. The proposed splitting of the training and inference tasks is interesting, and seems to perform quite well.
3. The generated molecules look reasonable and have relatively strong Vina scores, and the other chemical properties are acceptable as well. Lots of baselines are compared to, and IBEX appears to outperform them by a somewhat significant margin.

**Weaknesses:**

In general, I think the weaknesses of this paper come from the definitions that were used to define the three tasks. The de novo task is realistic, but the SC and SH tasks are not very realistic in my opinion.  First, SC would be somewhat close to a lead optimization task, but usually in lead optimization you already have side chains and you want to modify or expand them to increase activity. Second, SH is not realistic because scaffold hopping involves searching for both a new scaffold and new sidechains, not just a new scaffold. Setting the sidechains in place in particular is not representative of a scaffold hopping project. Because of these definitions used (which is understandable given the difficulty of setting up a apples-to-apples comparison), I don't think the result that SH is the best datasource to use is very meaningful. This calls the theoretical analysis part of this paper into question, but I think the empirical results are still interesting. Other weaknesses include:

1. Novelty is claimed in the physics refinement process, but I don't see how this is different from a standard force field minimization (which can include having a rigid ligand).
2. As far as I can tell, the physics refinement process is not applied to the baselines. This makes for an unfair comparison, because physical refinement is used for ligands generated by IBEX but not baselines.
3. Fig 1 is very confusing and looks AI generated... I would consider removing this or remaking it from scratch

**Questions:**

1. Why not use a standard forcefield for the physics refinement process, instead of a custom function?
2. I might be looking at the wrong place, but I don't see many details about how the trained model is transferred to the de novo setting from SH?

---

### Official Review · Reviewer_uhCo · 2025-10-31

**Soundness:** 3
**Presentation:** 2
**Contribution:** 3
**Rating:** 4
**Confidence:** 2

**Summary:**

This paper proposes a novel framework for 3D molecular generation using the PAC-Bayes information bottleneck framework to address the challenge of limited protein-ligand complex data. The authors introduce IBEX, a decoupled generation framework that leverages scaffold hopping to structure its latent representation of chemical space and applies it to de novo generation in a zero-shot transfer setting.

**Strengths:**

1. Innovative Information Bottleneck Approach: The paper introduces a novel application of the PAC-Bayes information bottleneck framework to molecular generation tasks, providing a theoretical basis for understanding the information density of different generation paradigms.

2. The proposed framework fully utilizes prior chemical information and achieves good results in terms of docking success rates, binding energies and so on, demonstrating its effectiveness in generating high-quality molecules under limited data conditions.

**Weaknesses:**

1. The decoupled approach and the use of multiple modules (e.g., scaffold hopping, physical refinement) may increase the complexity of the overall framework, potentially making it harder to implement and fine-tune compared to more integrated methods.

2. The method appears to be a mere adjustment of the training approach based on TargetDiff. It would be more convincing if its performance could be validated across other diffusion architectures as well (e.g. molcraft or else), which might better substantiate its effectiveness.

3. The poor Vina score, particularly in the absence of Physics-guided Position Refinement, is quite concerning. Given that Vina Dock involves re-optimization of conformations, does this suggest that the quality of the conformations directly generated by the method is subpar? Considering the array of evaluation metrics available for Structure-Based Drug Design (SBDD) tasks, it would be beneficial to incorporate additional metrics that specifically assess the rationality and quality of the generated conformations.

**Questions:**

Refer to the weakness part. I am intrigued by the innovative aspects of the method and am willing to reconsider my evaluation based on the feedback from other reviewers and the authors' response.

**Details Of Ethics Concerns:**

NA.

---

### Note · Authors · 2025-11-14

I have read and agree with the venue's withdrawal policy on behalf of myself and my co-authors.